


# Gravity waves above the Northern Atlantic and Europe during streamer events using ADM-Aeolus

Sabine Wüst[1], Lisa Küchelbacher[1], Franziska Trinkl[2], Michael Bittner[1,3]

[1] Erdbeobachtungszentrum, Deutsches Zentrum für Luft- und Raumfahrt Oberpfaffenhofen, 82234 Wessling, Germany

[2] formerly at Erdbeobachtungszentrum, Deutsches Zentrum für Luft- und Raumfahrt Oberpfaffenhofen, 82234 Wessling, Germany, now at Institute of Meteorology and Climate Research - Atmospheric Trace Gases and Remote Sensing, Karlsruhe Institute of Technology, 76131 Karlsruhe, Germany

[3] Institut für Physik, Universität Augsburg, 86159 Augsburg, Germany

*Correspondence to*: Sabine Wüst (sabine.wuest@dlr.de)

**Abstract**

Information about the energy density of gravity waves (GWs) is crucial for improving atmosphere models. So far, most space-based studies report on the potential energy, $E_{pot}$, of GWs, as temperature measurements from satellite are more common.

We use ADM-Aeolus (Atmospheric Dynamics Mission) wind data to derive a lower limit of the kinetic energy density, $E_{kin}$, of GWs above the Northern Atlantic and Europe. Aeolus on ADM, ESA's fourth Earth Explorer Mission, was the first Doppler wind lidar in space and measured vertical profiles of the horizontal line-of-sight wind from the ground to the lower stratosphere (20 – 30 km) between 2018 and 2023. With a vertical resolution of 0.25–2 km, Aeolus measurements are in principle well suited for the analysis of GWs. However, the data quality is a challenge for such analyses, as the accuracy of the data is in the

range of typical GW amplitudes in the tropo- and stratosphere.

In this study, we derive daily resolved time series of the lower limit of the $E_{kin}$, called $E_{kin,low}$, before, during and after two so-called streamer events above the Northern Atlantic and Europe. Streamers are large-scale tongue-like structures of meridionally deflected air masses, which are due to enhanced planetary wave activity. They are linked to vertical shear of horizontal wind and a pressure system, two possible GW generation mechanisms. We find that there is a temporal coincidence

between the daily averaged $E_{kin,low}$ and occurrence of the streamer events. The results indicate, that the derivation of GW signals based on Aeolus data is possible. However, we collected about 100 profiles to statistically reduce the uncertainty of the daily averaged $E_{kin,low}$. Compared to non-satellite measurements those daily averaged values are at the upper border.





## 1    Introduction

Gravity waves (GWs) transport energy and momentum over large distances in the atmosphere without a net mass transport.
Primary GW are mostly excited in the lower atmosphere (e.g. Fritts and Alexander, 2003, Pramitha et al. 2015 for a case study) and even though there exist wave phenomena with larger periods and wavelengths in the atmosphere, GWs dominate atmospheric dynamics especially above 75 km height (Houghton, 2002). In the absence of a background wind, energy and momentum are conserved quantities; if the background wind is not zero, the pseudo-energy and -momentum, which are can be derived from the energy and some wave parameters, are conserved (Nappo, 2013). Deviations indicate regions in the
atmosphere, where GW irreversibly influence temperature (through (pseudo)energy deposition) and wind (through (pseudo)momentum deposition) while they lead to reversible changes elsewhere. Therefore, two of the main questions when investigating GW are "how much (pseudo-)energy and (pseudo-)momentum are transported?" and "where are they deposited?". Answers to those questions and a proper parameterization is one key to improve weather and climate models (Eichinger et al. 2020, Alexander et al. 2010).


Information about kinetic and potential energy are often provided as densities in literature (e.g. Ern et al., 2018; Rauthe et al., 2008; Wüst et al., 2016), so energy per unit mass (in J/kg) or per unit volume (in J/m$^3$). Those quantities are calculated as follows:

$$E_{kin} = \frac{1}{2}(u'^2 + v'^2 + w'^2)$$

45                                                                                                          (1)

where $(u', v', w')$ represent the wind fluctuations due to GWs averaged over one phase, and

$$E_{pot} = \frac{1}{2}\frac{g^2}{N^2}\overline{\left(\frac{T'}{\bar{T}}\right)^2}$$

(2)

where $g$ is the acceleration of gravity, $N$ is the Brunt-Väisälä frequency, and $T'$ is the temperature fluctuation, and $\bar{T}$ the
background temperature. The overbar denotes the average over one period or multiples of it.

Information about the momentum are usually provided in terms of vertical flux of horizontal momentum which is

$$\left(F_{px}, F_{py}\right) = \bar{\rho}\left(1 - \frac{f^2}{\hat{\omega}^2}\right)\left(\overline{u'w'}, \overline{v'w'}\right)$$

(3)

$F_{px}$ and $F_{py}$ are the zonal and meridional momentum flux components, $\bar{\rho}$ is the atmospheric background density, $f$ is the
Coriolis parameter and $\hat{\omega}$ the intrinsic frequency ($\hat{\omega} = \omega - u \cdot k - v \cdot l$ with $k$ and $l$ the zonal and meridional wave numbers and $u$ and $v$ the zonal and meridional background wind).

From the equations (1) – (3), it is apparent, that an ideal GW satellite mission would measure both, temperature and wind, in three dimensions. The data would need to be decomposed into background and GW-induced fluctuations. From the





fluctuations, the wave vector and the period could be derived. However, for parts of the GW spectrum equation (3) can by

simplified to a version without wind information using linear polarization equations (Ern et al., 2018).

Temperature and wind have been measured from satellite for many years. However, temperature information is available much more frequent than wind measurements. A comprehensive overview about the past satellite-based wind missions in the upper mesosphere and lower thermosphere (UMLT) is given in the introduction of Dhadly et al. (2021). By the end of 2022, three

wind missions were in orbit. They all use the principle of Doppler to derive horizontal wind information but they target different species and therefore different altitudes. Two systems are passive and use different airglow emissions and only one is active. The two passive ones were MIGHTI (Michelson Interferometer for Global High-resolution Thermospheric Imaging) on the NASA Ionospheric Connection Explorer (ICON) mission and the TIMED Doppler Interferometer (TIDI, Niciejewski et al., 2006) on the Thermosphere Ionosphere Mesosphere Energetics and Dynamics (TIMED) satellite. MIGHTI delivered vertical

profiles of the horizontal wind (height ranges vary, day: 90 – 300 km, night: 90 – 105 km and 200 – 300 km) from 2019 until the end of 2022. It used a doppler asymmetric spatial heterodyne spectrometer which measures the Doppler shift of the oxygen red and green airglow line at 630.0 and 557.7 nm (Englert et al., 2017). TIDI started its operation more than 20 years ago in 2002. It uses a Fabry Perot interferometer to measure the Doppler shift of individual emission features of the $O_2$ (0,0) airglow band. From these shifts, horizontal winds between 70 and 120 km during the day and 80 and 105 km at night can be derived

(Dhadly et al., 2021). Aeolus on ADM (Atmospheric Dynamics Mission, Tan et al., 2008), ESA's fourth Earth Explorer Mission, was an active mission and the first Doppler wind lidar in space. Since 2018, it measured vertical profiles of the horizontal line-of-sight (hlos) wind from ground to the lower stratosphere (20 – 30 km) (Reitebuch et al., 2020; Tan et al., 2008). It carried the Atmospheric LAser Doppler INstrument (ALADIN), which emitted in the UV range (354.8 nm). The Doppler shift of the backscattered radiation (Rayleigh and Mie) was analysed. Aeolus stopped operation on April, 30[th] 2023

(https://www.eumetsat.int/end-nominal-aeolus-mission-operations, date of access: July, 3[rd] 2023).
Aeolus wind measurements in principle enable the global derivation of $E_{kin}$. These data has a vertical resolution of 0.25–2 km, which is well-suited for the analysis of GWs. Especially, a lower limit for the kinetic energy density, denoted as $E_{kin,low}$, can be derived. In the upper troposphere / lower stratosphere (UTLS), GWs typically show amplitudes of 5–10 m/s at maximum (e.g., Dutta et al., 2017; Kramer et al., 2015). Challenging is the accuracy of Aeolus, which is lower than originally planned

for and now in the same order of magnitude as typical GW fluctuations.

This study concentrates on specific dynamical situations called streamers in the upper troposphere / lower stratosphere. Streamers are large-scale tongue-like structures of meridionally deflected air masses (Hocke et al., 2017; Krüger et al., 2005; Offermann et al., 1999); they are linked to enhanced or breaking planetary waves (PWs). PWs are associated with the formation

of pressure systems and influence the position of the tropospheric jet. Enhanced or breaking PWs lead to strong shears of the horizontal wind. Pressure systems, wind shears, but also the jet are possible sources of GWs (e.g., Plougonven and Zhang, 2014; Zülicke and Peters, 2008; Fritts and Nastrom, 1992).



On a case study basis, we investigate whether enhanced $E_{kin,low}$ can be observed during two pronounced streamer events
above the Atlantic in February and November 2020 based on ADM-Aeolus measurements. Of all the space-based wind sensors
listed above, ADM-Aeolus is the only one that addresses the region where streamers exist and therefore the region of potential
GW generation.

The manuscript is structured as follows. The selection and identification of the streamer events which were analysed with
respect to GW is given in section 2. Section 3 comprises a description of the data basis, which consist of ADM-Aeolus wind
data. In section 4, it is explained which ADM-Aeolus data we accepted for analysis, how we extracted GW signatures and
calculated $E_{kin,low}$ as well as its error. In section 5, the temporal evolution of $E_{kin,low}$ is presented before, during and after two
pronounced streamer events. In section 6, the results are discussed. The manuscript ends with a summary and conclusion
(section 7).



## 2    Dynamical situation

As mentioned in the introduction, we focus on streamer events as a generation mechanism for GWs (at their flanks due to strong wind shear but also convective GW since a streamer event is linked to a strong anticyclone). When using the term "streamer", one has to provide some further information since this expression is not uniquely defined as pointed out by Krüger et al. (2005) in their introduction. Those authors provide a comprehensive overview about the research on streamers, their effect on mixing, the different definitions, etc. We refer here to large-scale tongue-like structures of meridionally deflected air masses as they are described by Offermann et al. (1999).

Those streamer events can be separated into tropical-subtropical streamers, which transport air of low into mid latitudes, and polar vortex streamers leading to a mixing of polar air into mid latitudes. Krüger et al. (2005) published a climatology of both streamer types based on 10-year model runs: those events mainly occur during October and May (on the Northern hemisphere) over East Asia and the Atlantic.

Streamers can be traced by ozone, as it has a comparatively long life-time in the lower stratosphere. The reason for streamer events is planetary waves, which are the main drivers of the extratropical circulation. They lead to an irreversible mixing of air masses between the equatorial and polar region (e.g., McIntyre & Palmer 1983, Polvani & Plumb 1992). Normally, streamers can be observed for some days.

The identification of streamer events is based on global maps of total ozone column measurements (TO3), which are available as a service by DLR (https://atmos.eoc.dlr.de/app/calendar). TO3 is retrieved by the Tropospheric Monitoring Instrument (TROPOMI) on the Sentinel 5 Precursor (S5P) satellite. Whenever no data by TROPOMI/S5P is available, TO3 measurements of the Global Ozone Monitoring Experiment-2 (GOME-2) on the Metop series of satellites are considered. The instruments are nadir-viewing on a near-polar sun-synchronous orbit. TROPOMI/S5P was launched in 2017 and has a spatial resolution of 7 x 7 km$^2$ with a daily global coverage and a repeat cycle of 17 days (Veefkind et al. 2012). Details on TO3 by TROPOMI/S5P are given by Spurr et al. (2022). The TO3 retrieval is based on the processor of the previous GOME instrument: GOME-2 on Metop-AB was launched in 2006. It has a spatial resolution of 80 x 40 km$^2$ and almost a daily global coverage with a repeat cycle of 29 days. See Munro et al. (2006) and Munro et al. (2016) for an overview of the instrument and data processing. Details of the GOME-2 retrieval algorithm can be found in Loyola et al (2011).

In this study, we focus on tropical-subtropical streamers over the Northern Atlantic. The events are identified manually considering the daily TO3 global maps from January 2020 to March 2021. We found three events (approximately 6. – 11. February 2020, 4. – 8. September 2020, and 1. – 8. November 2020) which are, from our perspective, strongest in their evolution, i.e., they have a comparable large spatial size, high intensity (low TO3 concentration), the TO3 low air masses are irreversibly mixed into the surrounding atmosphere (cut-off) and they can definitely be distinguished from other dynamical mixing and vortices.





Figure 1 shows a streamer event over the Northern Atlantic in November 2020. The flanks are comparatively parallel to the longitudes, so it has a strong meridional structure. It reaches regions latitudes of 70°N. A smaller streamer (which is not

considered in this study) can be detected over western North America. There are also ozone-poor air masses above eastern Europe.

Figure 2 shows a streamer event over the Northern Atlantic in February 2020. It is characterized by a diagonal spatial extension from the Sargasso Sea in the south-west reaching the southern parts of Ireland in the north-east at approximately 50°N. Compared to the streamer in November 2020 this one stands out more from its surroundings. This is due to the fact that the

ozone content of the surrounding air is higher in February 2020 than in November 2020, while the ozone concentration within the streamer is roughly the same.

The streamer observed in November 2020 is not shown here, as the Aeolus data coverage is not good enough for further analysis (see also section 3).


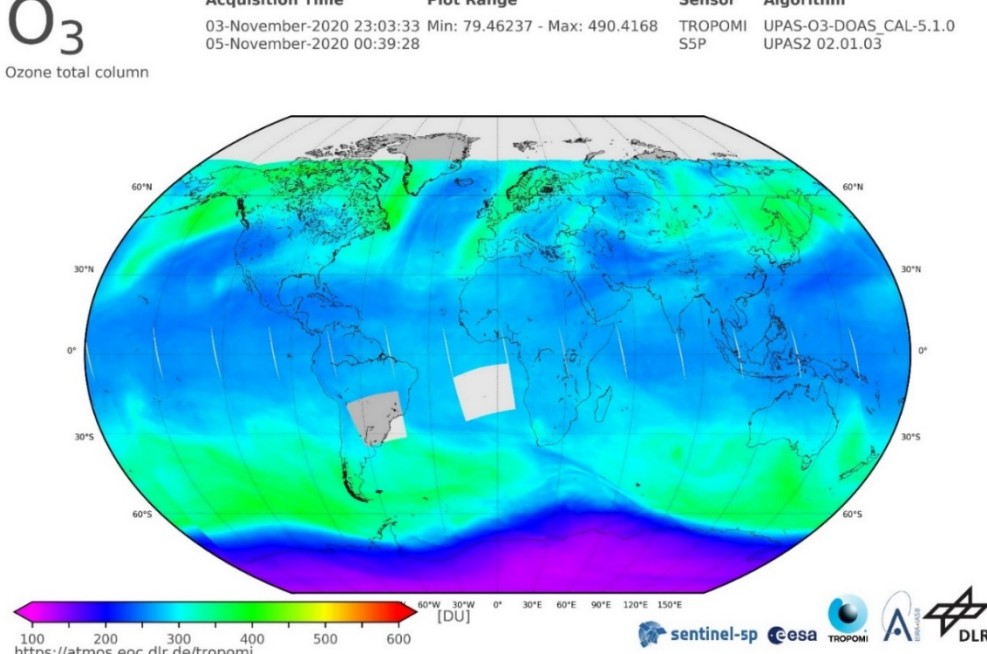

**Figure 1 The TO3 measurements taken by TROPOMI/S5P on Nov., 4th 2020 indicate a large streamer event. The starting data cannot be clearly specified due to data gaps. On Nov. 1st a slight signature was already visible above the eastern coast of North**
**America. The streamer moved east ward with time. It was most pronounced above the Northern Atlantic on Nov., 4th. On Nov. 8th, it was not visible any more. Source: DLR, CC-BY 3.0**



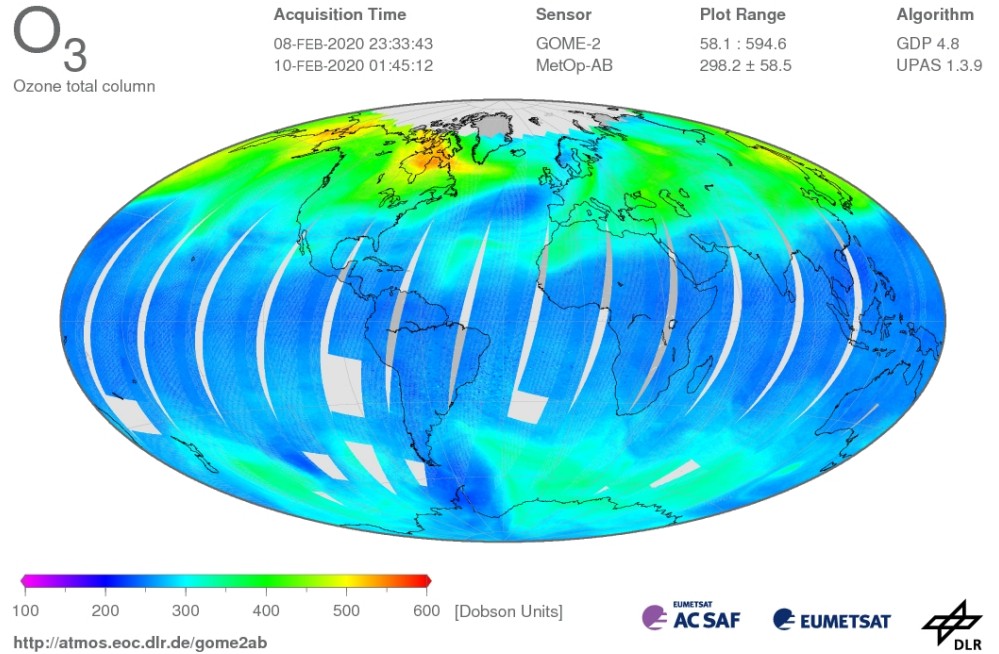

**Figure 2 The TO3 measurements taken by GOME-2/MetOp-AB on Feb., 9[th] 2020 indicate a large streamer event. It started to evolve on Feb., 6[th] in front of the Eastern coast of Northern America and moved westward with time. It was most pronounced above the Northern Atlantic on Feb., 9[th]. On Feb. 12[th], it was not visible any more. Source: DLR, CC-BY 3.0**




## 3    Data basis

Detailed information about the Aeolus instrument is given on the ESA homepage https://www.esa.int/Applications/Observing_the_Earth/FutureEO/Aeolus/Documents_publications (last access: 16[th] September 2022). Here, only the very basics are summarized. ADM-Aeolus was launched on August, 22[nd] 2018. It carried a Doppler wind lidar (ALADIN, Atmospheric LAser Doppler INstrument), which emitted in the UV range (354.8 nm). The backscattered radiation was collected by a telescope, its Doppler shift was derived and analysed. ADM-Aeolus measured both,

the Rayleigh and Mie backscattering. The first one originates from molecules, the second one from particles.

Aeolus data are provided as dbl (Data BLock) files from ESA. They contain amongst others vertical profiles of the hlos (horizontal line of sight) wind velocity. We downloaded the level 2B (L2B) data and added a variable which contains the hlos wind corrected for the satellite observation geometry. We selected the variables which we needed for our analyses with the CODA software in python. Finally, we converted everything to ncdf files. The list of the variables in the ncdf files is given in

Table 1.

A correction of the sign of the wind measurements is necessary, since the hlos wind is given relative to the satellite. The instrument looks to the right side of the satellite with respect to the flight direction. Wind which blows away (towards) from the instrument has a positive (negative) sign. Therefore, distinguishing between ascending and descending mode is necessary for providing the hlos wind independent of the satellite observation geometry. This can be done by using the variable

'los_azimuth', which informs about the horizontal position of the satellite relative to the target in degrees. The target is in the centre of the coordinate system and the azimuth is the angle between the vector pointing from the target to the North and from the target to the satellite. If the azimuth angle is larger than 180°, the satellite is west of the target and therefore in its ascending orbit branch and vice versa. In atmospheric physics, a wind to the East (West) has a positive (negative) sign. A positive hlos wind is a wind to the East in the ascending branch and to the West in the descending branch. So, in order to become independent

of the satellite observation geometry, the sign of the hlos wind must be changed, if the azimuth angle is smaller than 180°.

Hlos wind is available as four observational products: Rayleigh clear, Rayleigh cloudy, Mie clear, and Mie cloudy. For the Rayleigh wind measurements, which represent the majority of the wind measurements, 30 individual measurements are averaged; therefore, each hlos wind value is the horizontal average over 86.4 km (Martin et al., 2021).

**Table 1 Overview table of auxiliary data files of Aeolus data.**

| Variable Names | Variable Description | Variable Units |
|---|---|---|
| alt | Height of the wind measurement | m |
| hlos | Hlos wind | cm/s |
| hlos_corrected | Corrected hlos wind dependant on ascending or descending mode | cm/s |





| hlos_error | Error estimate reported by the Rayleigh processing algorithm (defined in RD7) | cm/s |
|---|---|---|
| lat | Latitude of the wind measurement | degrees |
| lon | Longitude of the wind measurement | degrees |
| los_azi | Azimuth of the target-to-satellite pointing vector measured clockwise from north. | degrees |
| observation_type | Information if the profile is cloudy or clear sky. Values range from 0 to 2 which mean<br>0 for initialization purposes only<br>1 cloudy<br>2 clear | |
| start_of_obs_datetime | Date and time of first measurement used for wind result | seconds since 2020-10-08 09:09:47.0260 30976 |
| validity_flag | Indicates if data is valid (1) or invalid (0) | |

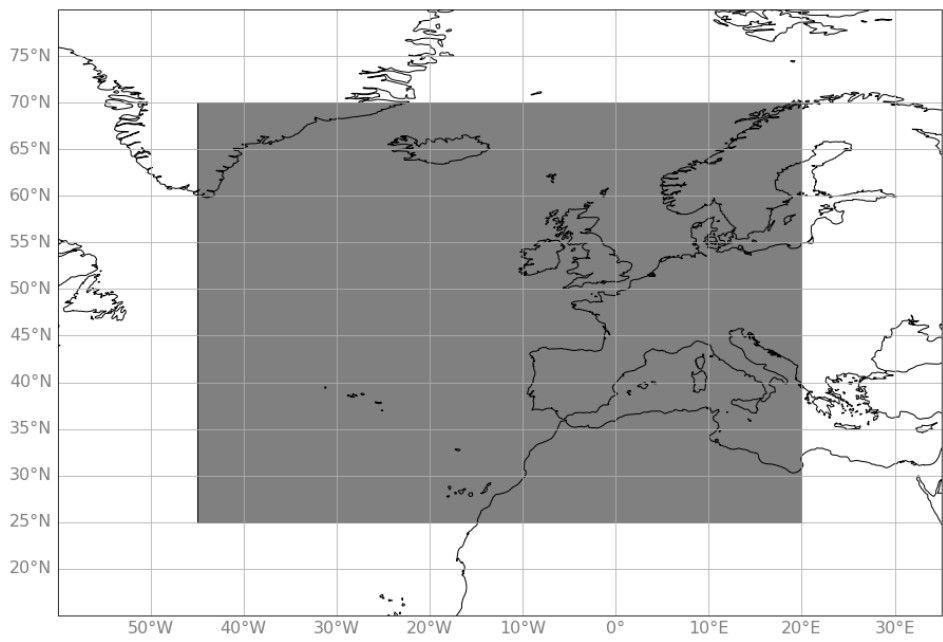

**Figure 3 Map showing the region investigated for GW. Streamer 1 was observed in this region during the period of November 3 to November 7, 2020, streamer 2 from February 8 to February 11, 2020.**




For our analyses, we used only Rayleigh clear wind measurements, which were marked as valid (variable 'validity_flag' equal to 1). According to the geographical position of the streamer events, which was discussed in the previous section, the spatial focus for the investigation of GW in this manuscript is on 25–70°N and 45°W–20°E (see Figure 3). For Aeolus, different processor baselines are available, which cover different observation periods. In this study, we used data referring to the year 2020, this is baseline 11 (2B11). A consistent reprocessed data set covering the Aeolus whole observation time was not available for the time of our calculations.

For our purpose, only the statistical error of the hlos is of importance as long as the systematic one stays approximately constant with height, as we will explain later. Unfortunately, ESA does not provide individual information about the height of the different error types, only an integrated value (hlos_error, see Table 1) is given. Martin et al. (2021) separated this error into a systematic and a statistical one. In their study, the authors used Rayleigh winds with an estimated error of 6 m/s at maximum; they refer to 2B02–2B07 (September 2018 until December 2019), so to earlier baselines than we do. The absolute bias averaged over the analysed time period is ca. 2 m/s, but it depends on the data set used for validation and on the ascending or descending orbit (see table 1 of Martin et al. (2021)). To some extent, this is due to the six different processor baselines. Furthermore, there were several updates of the calibration files. The estimated Aeolus instrumental error is given by 4.0 – 4.4 m/s for Rayleigh winds (see table 2 of Martin et al. (2021)). It shows a temporal variation, which mainly depends on the laser output energy. Ratynski et al. (2023) compared Aeolus to radiosonde and ground-based lidar data during a longer time period (September 2018 – January 2022, which means 2B02 to 2B13) than Martin et al. (2021) but only at two stations. Furthermore, they did not apply any hlos error threshold. Those authors report an averaged systematic error of -0.92 m/s and -0.79 m/s and a mean random error of 6.49 m/s and 5.37 m/s for lidar and radiosondes, respectively. They state that the bias correction of the Aeolus data which took place around mid-April 2020 did not affect the random error.

Aeolus 2B data are delivered on 24 height bins. Data at high altitude levels are not available for each Aeolus profile, data at low altitudes suffer from slightly larger errors than other heights (investigated for November 2020, 25°–70°N, 0°–20°E). The histogram of the error of all height bins shows a structure with three maxima (see Figure 4). From this plot, one can conclude that accepting an error larger than ca. 4.5 m/s will not enlarge the data basis significantly. In order to find a compromise between profiles which are as long but also as accurate as possible, we decided for two quality criteria: the profiles must cover all height bins 1–21 (with height level 1 the lowest altitude) and the accuracy has to be better than 4.5 m/s at each height bin. As mentioned above, there exist three pronounced streamer events during January 2020 to March 2021. In this study, the events in February and November 2020 are analysed. For the streamer in September 2020, there is not enough Aeolus data available to meet our quality criteria.

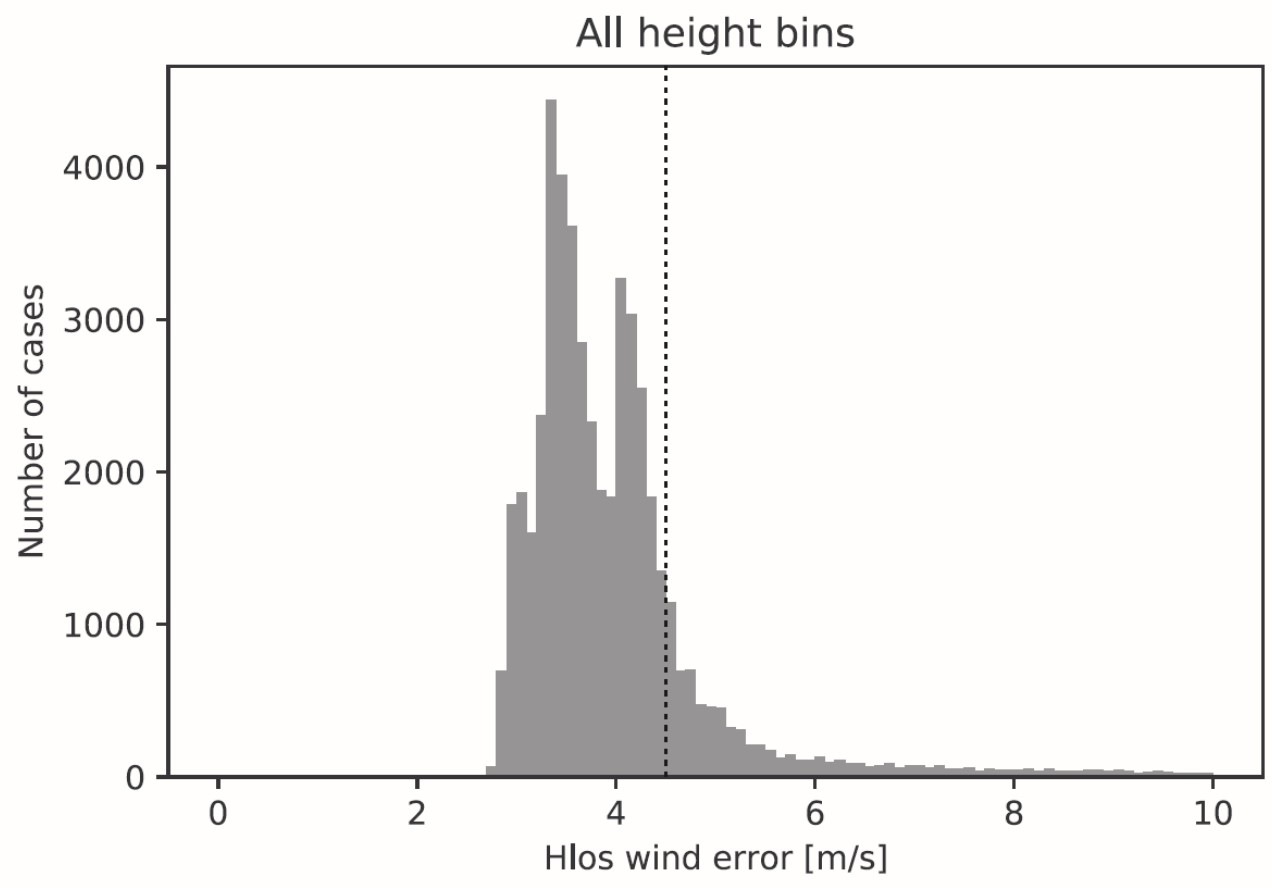


**Figure 4 The histogram for the whole November 2020 shows a characteristic structure with three maxima at ca. 2.9 m/s, 3.3 m/s and 4.1 m/s. None of the profiles in the investigated geographical area (25°–70°N, 0°–20°E) has a better accuracy than ca. 2.7 m/s. These values refer to all height bins, so 1 – 24. The vertical dashed line marks an error of 4.5 m/s.**





## 4    Analysis

We rely here on cubic splines for the approximation of the atmospheric background, so on piecewise third-order polynomials which are stitched together at the spline sampling points following some mathematical specifications. A cubic spline is adapted individually to each vertical wind profile and subtracted from it. The residuals are analysed further.

The distance of the spline sampling points determines the sensitivity of the spline: according to the Nyquist sampling theorem, a signal with a vertical wavelength of x km, must be sampled at least every x/2 km in order to resolve it. Transferred to the

spline, this means that a spline sampling point must be set every x/2 km. An alternative formulation is: a spline with sampling points every x/2 km is sensitive to wavelengths of x km and longer. The residuals include only signals with a vertical wavelength of x km at maximum.

There exist two main challenges for splines. The first one, which is common to all adaption techniques, is the insufficient approximation of extrema in the background, e.g., the wind maximum in the tropopause. In this case, the spline is too smooth

and its subtraction from the original data leads to an artificial signature in the residuals. Secondly, a spline can generate artificial oscillations (at the beginning and the end of a vertical profile or also over the whole height range, if the vertical wavelength to which the spline is sensitive is approximately equal to the vertical wavelength of the GW which is present in the data, see Wüst et al. (2017)). These undesired effects are less pronounced when using the repeating spline approach, i.e., the profile is adapted by splines with the same distance between their sampling points but with varying starting points. Their mean is used as final

approximation. This approach is introduced and discussed especially for the derivation of GW signals in Wüst et al. (2017). It is applied here.

The profiles chosen for the GW analysis due to our quality criteria have a length of approximately 16 km and a vertical resolution of ca. 0.8–1.0 km. Therefore, the minimal vertical wavelength to which the spline is sensitive and therefore the maximal vertical wavelength of the residuals can range between ca. 2 km (Nyquist criterion) and 16 km. Since GW with a

short vertical wavelength also have small amplitudes in most cases and since the error of Aeolus is relatively large for the investigation of GW in the troposphere, the choice of 2 km as maximal wavelength in the residuals will probably not deliver useful results. The same holds for the other extremum: the choice of a maximal wavelength of 16 km will lead to a coarse approximation of the tropopause wind extremum and therefore to relatively strong artificial signatures in the residuals at the

tropopause height. That is why we decide for three different and more moderate maximal vertical wavelengths (5 km, 7.5 km and 10 km). The splines are adapted and the residuals are calculated. For the calculation of the density of kinetic energy $E_{kin}$ knowledge about the three-dimensional wind variations generated is necessary (see equation (1)).

As mentioned above, Aeolus measures the line-of-sight wind. Due to the orientation of the instrument and the satellite orbit

this is to a large extent the zonal wind. Using L2B data, one can only calculate a lower bound for $E_{kin}$, which we call $E_{kin,low}$. $E_{kin,low}$ is derived for every height bin. Since the profiles cover different height ranges, $E_{kin,low}$ is summed up and divided by





the number of values. We call the result mean $E_{kin,low}$. It has the unit $\frac{J}{kg}$. The algorithm is sketched in Figure 5. Based on this algorithm, GW hotspots in place or time, so maxima of $E_{kin,low}$ over a certain area or in a specific time period as defined by the geographical and temporal parameters at the beginning of the algorithm can be identified.


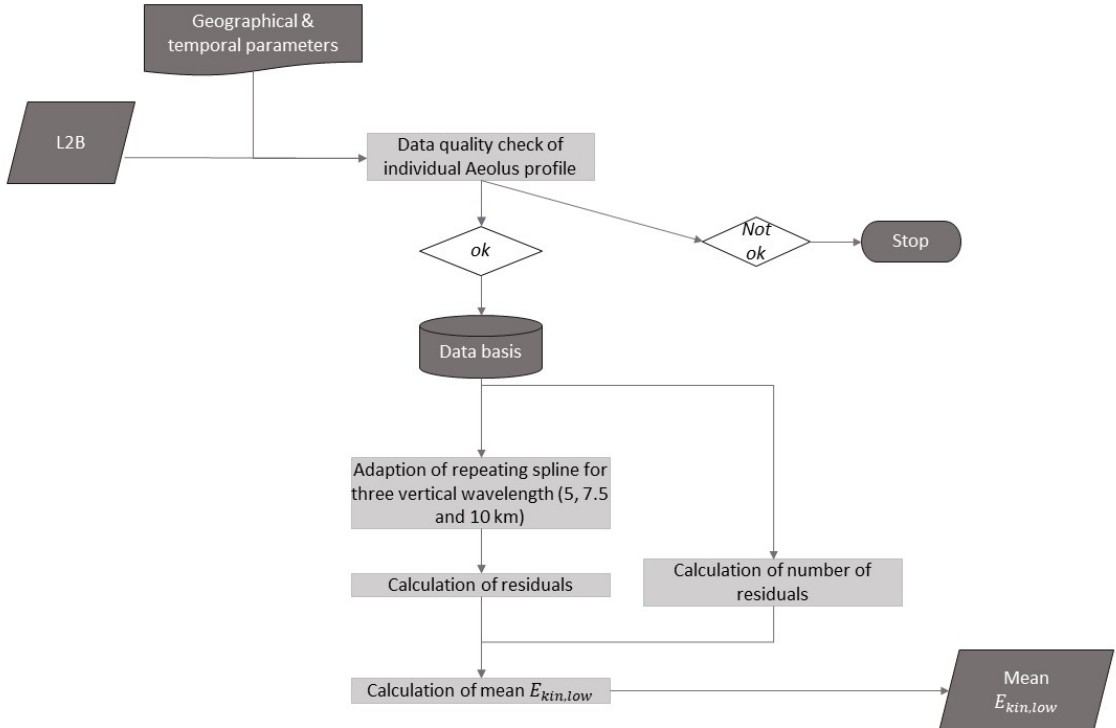

**Figure 5 Scheme of the algorithm for the derivation of the mean $E_{kin,low}$ from ADM-Aeolus L2B wind measurements.**


An important point in this context is the estimation of the error of $E_{kin,low}$. This can be done in a similar way as already demonstrated by Kramer et al. (2016) in their appendix A who used a (non-repeating) cubic spline in order to derive GW signatures from radiosonde-based measurements in the lower stratosphere. Those authors generated a data basis with simulated GW signatures (five different starting amplitudes (0.5 K, 1.0 K, …, 2.5 K), five different values used to increase the amplitudes

linear with height (0.05, 0.01, …, 0.25) from 0 – 29.7 km height, 13 different wavelengths (1.0 km, 1.5 km, …, 7 km) and eight different phases (π/8, π/7, …, π)). They arbitrarily constructed five GW, superimposed them on a realistic atmospheric background and detrended the generated profiles with a spline sensitive to wavelengths of 7 km and longer. So, in principle this spline should be able to remove all GW signatures from the background. They repeated this approach about 1000 times,





calculated the difference between the original background and the retrieved one and derived the mean over these differences
depending on height in order to estimate the error induced by the spline.

We do some adaptions for our error calculation; the algorithm is sketched in Figure 6. Since the vertical resolution of Aeolus
data is coarser compared to radiosondes and therefore less GWs are visible in Aeolus profiles, we superimpose only three
oscillations with maximal wavelengths between 2.0 km and 10 km. The height range we investigate covers 1–17 km, the
vertical resolution is 800 m. We assume, that the amplitudes (here in m/s instead of K as in Kramer et al. (2016)) can grow
linearly between 0.05 and 1.95 over the height range. As shown later in section 5.1 (see Figure 7), this is a valid assumption
overall. As background data, we use CIRA. Since the vertical resolution of CIRA is different to the vertical resolution of
Aeolus, a spline is used to adapt the resolution of CIRA to the one of Aeolus. Since we are not only interested in the error due
to the spline approximation, but would also like to include the accuracy of Aeolus, we add a value for the measurement
uncertainty to the background and the superimposed GW signatures. We accept a maximal error of 4.5 m/s. As mentioned
above, Martin et al. (2021) showed for Aeolus data with an error of 6 m/s and better that ca. 2/3 of the accuracy is due to
random processes and ca. 1/3 can be attributed to a bias. A systematic bias approximately constant with height is not important
for the GW derivation since it will be taken out through the detrending procedure. We check the distribution of the Aeolus
accuracy with height for November 2020. Although it shows some height-dependent variation (in general lower in the lower
part and higher in the upper), it cannot be modelled by a simple function such as a linear one. Therefore, we calculate the mean
value over the used height bins (1–21), which is ca. 3.7 m/s, and its standard deviation (ca. 0.3 m/s). Since we already use
Aeolus data of relatively high quality, we make the conservative assumption that 1 m/s of the 3.7 m/s is caused by systematic
effects and derive normally distributed values based on a mean of 2.7 m/s and the standard deviation mentioned above. For
each height, the algorithm arbitrarily chooses the accuracy from the normally distributed values and the sign.

Then, we detrend the profile with the repeating spline approach which is sensitive to 10 km at minimum and calculate the
residuals, which we compare to the original GW signatures. We repeat this process 100 times for different latitudes (40° N,
50° N, and 60° N) and use these values in order to calculate an error for $E_{kin,low}$.





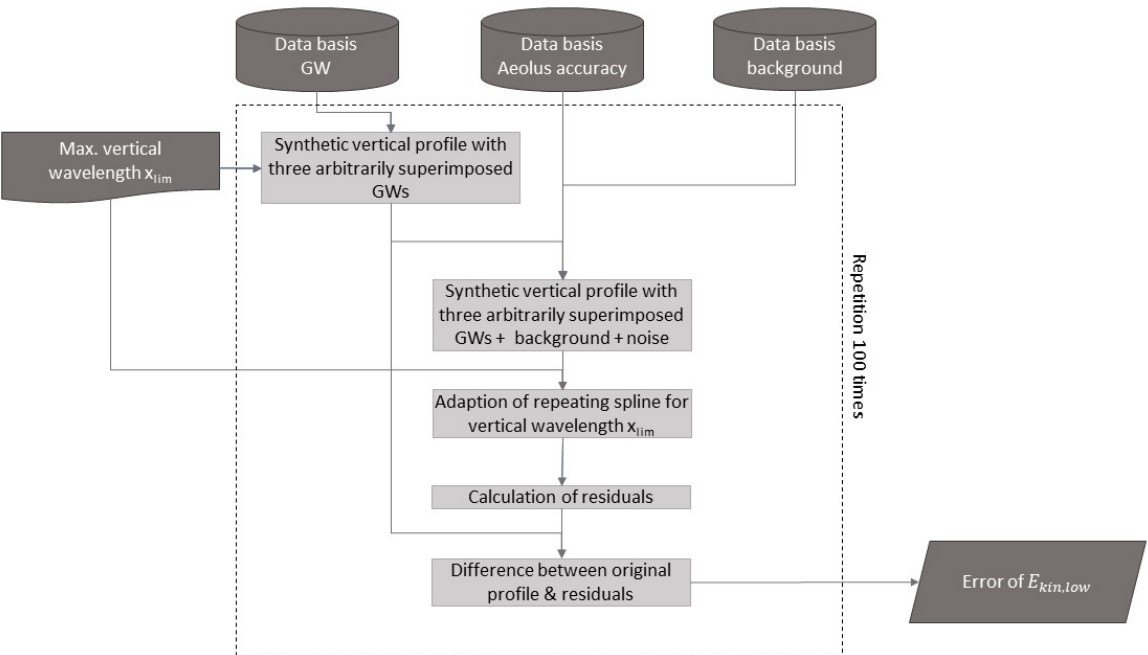

**Figure 6 Scheme of the algorithm for the error estimation of $E_{kin,low}$ from ADM-Aeolus L2B wind measurements.**





# 5    Results

We start with the event in November 2020, which shows a more meridional structure than the streamer in February 2020 (see section 2). The November streamer is referred to as streamer 1 in the following. Accordingly, the February streamer is denoted

as streamer 2.

Figure 7 shows the vertical profile of the absolute residuals averaged over time for different maximal vertical wavelengths (2750 profiles, from 15[th] October to 29[th] November 2020, so including the time of occurrence of streamer 1 as well as approximately the two to three weeks before and after the event). The values of up to 5 m/s are in the same order of magnitude

as values reported in literature (e.g., by Kramer et al. (2015) or Nath et al. (2009)).

a)

b)



**Figure 7 Shown are the absolute residuals per height (a) averaged over 15th October – 29th November 2020 (2750 profiles). The colours refer to the different maximal vertical wavelengths (blue: 5 km, orange: 7.5 km, and green: 10 km). The profiles originate from the geographical area shown in Figure 2 (25–70°N / 45°W–20°E). Part b) splits the residuals into positive and negative ones.**

However, two limitations can be seen in Figure 7:

The first one is that the Aeolus error, which ranges between 2.8 m/s and 4.5 m/s, covers the range of mean residuals. This makes individual Aeolus profiles challenging to use for GW analysis. So either, only very pronounced events are investigated based on Aeolus data such as Banyard et al. (2021) showed it for wind perturbations larger than 10 m/s, or a larger amount of profiles is collected and the mean residuals are analysed. In the latter case, instead of the individual error, the mean error of the averaged residuals can be used, which is a factor $1/\sqrt{n}$ smaller than the individual error. We rely on the later solution here.

The second limitation is that there exists a maximum in the vertical profile of the mean absolute residuals between 7.5 km and 10 km height. Even though the repeating spline performs better than the non-repeating one, it cannot be excluded that we see an artificial signature, which should not be interpreted as GW signatures. To exclude this effect of the wind maximum, the residuals are only analysed below 7 km (tropospheric part) and above 11 km (stratospheric part) in the following.

For Figure 7, we calculated the temporal average and showed vertically resolved data. Now in Figure 8, we do it the other way around: we focus on vertically averaged data (split into tropospheric and stratospheric part as mentioned above) and show a time series with daily resolution.

Both, the tropospheric and the stratospheric parts (Figure 8a and b), are not characterized by a single pronounced maximum during mid-October to late November 2020, but there are a number of local maxima. Within the stratosphere or troposphere, the curves for the different wavelengths agree quite well. The agreement is slightly worse, if, for example, the mean $E_{kin,low}$ of GW with 10 km maximal vertical wavelength of the stratosphere (Figure 8a, green curve) is compared with the corresponding tropospheric curve (Figure 8b, green curve).

The mean $E_{kin,low}$ in the stratosphere shows a local maximum during 3rd to 5th November for all vertical wavelengths, while this is not the case for the troposphere. The stratospheric maximum is less pronounced for a maximal vertical wavelength of 5 km. Part c) of Figure 8 depicts only the maximum vertical wavelength of 10 km, this time shown with error bars. It is clear that the maximum during November 3rd to 5th is significant.

The values of the error bars come from the algorithm sketched in Figure 6. The error analysis described there was done for different latitudes since the background wind (from CIRA) varies to some extent depending on latitude. For $E_{kin,low}$ derived from an individual profile, an error of 1.9 – 2.4 J/kg or 2.2 J/kg on average can be expected (see Table 2). We use this mean value of 2.2 J/kg for the calculation of the error bars. Since our analysis relies on ca. 100 profiles per day, the mean error is approximately $\frac{2.2}{\sqrt{100}}$ J/kg.

For streamer 2, the development of the mean $E_{kin,low}$ is calculated from 1st to 28th February 2020 for a maximal vertical wavelength of 10 km and separated into the tropo- and the stratospheric part (Figure 9). Also in this case, both, the tropospheric





and the stratospheric $E_{kin,low}$, are not characterized by a single maximum during the occurrence of the streamer but by a number of local maxima. The stratosphere shows a pronounced and significant maximum on February 10th. For the $E_{kin,low}$ of the troposphere, this is not the case; here $E_{kin,low}$ is high from February 10th to 14th.


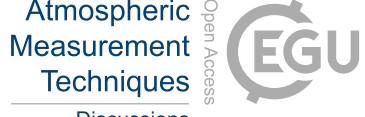

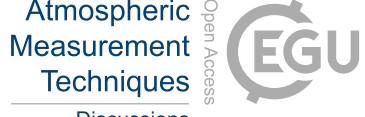



**Figure 8 Shown is development of the mean $E_{kin,low}$ over time. While subplot a) refers to the tropospheric part, the lower two figures**
**depict the stratospheric part. The colour code is the same as in Figure 6 (blue: 5 km, orange: 7.5 km, and green: 10 km maximal vertical wavelength). Part c) shows only the time series for the maximal vertical wavelength of 10 km. Additionally, error bars are included and the time period 3$^{rd}$ – 7$^{th}$ November 2020, where the streamer could be observed in TO3 measurements above the investigated region, is marked by the dotted lines.**

**Table 2 Provided is the error of $E_{kin,low}$ for one individual profile based on 100 arbitrarily generated and detrended vertical GW profiles. The latitudinal dependence is in the range of 11%.**

| Latitude | Mean error of $E_{kin, low}$ [J/kg] |
|---|---|
| 40°N | 2.4 |
| 50°N | 1.9 |
| 60°N | 2.2 |

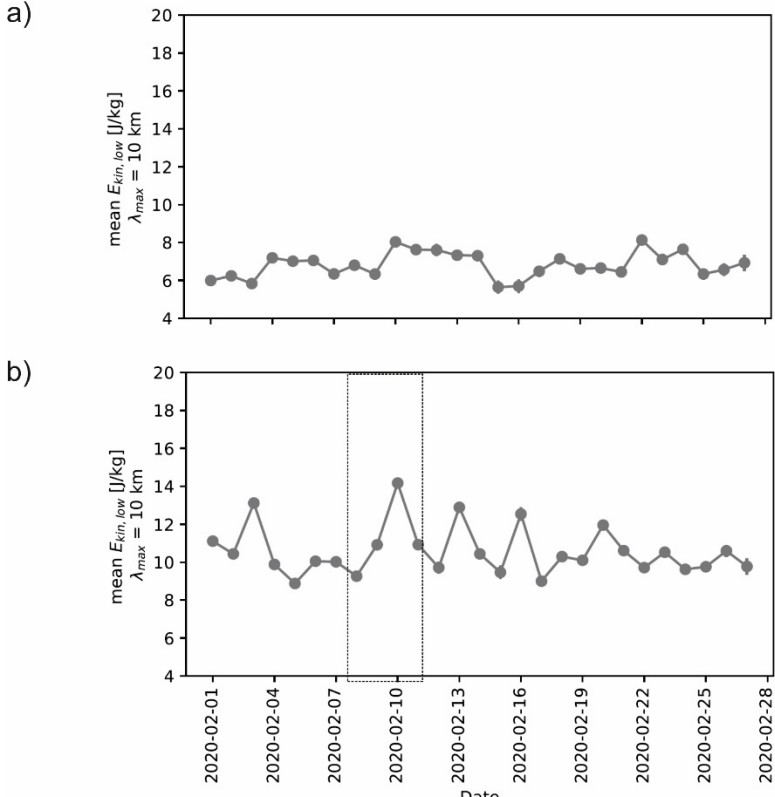

**Figure 9 Shown is development of the mean $E_{kin,low}$ over time for the troposphere (a) and the stratosphere (b). The dashed rectangle**
**shows the time period of streamer 2 in the investigated geographical region.**



## 6    Discussion

GWs can be generated by a number of processes, where air experiences a vertical momentum (which can be up- or downward). Additionally, GWs can originate due to unstable shears or geostrophic adjustment (accompanying frontal evolution, for example), or they can be the result of an interaction between other GWs (Fritts and Alexander, 2003). Several of those processes occur simultaneously with a streamer such as vertical movements, wind shear, and fronts. As shown in the section above, we focus on data of the lower stratosphere, which is a stably stratified part of the atmosphere. Besides the vertical momentum on an air parcel, a stable stratification of the surrounding atmosphere is the second prerequisite for the existence of GWs. So, in principle, it is not surprising to observe enhanced GW activity in the stratosphere during a streamer event. However, to our knowledge, such a study has not been published so far.

Additionally, we use ADM-Aeolus wind data here. As already mentioned before, the data quality of Aeolus is a challenge for the analysis of GWs as long as they are not very pronounced. While there exist publications, for example, on the validation of ADM-Aeolus data (e.g., Khaykin et al. (2020), Ratynski et al. (2023), Witschas et al. (2022)), to our knowledge only one article has been published, which focuses on the analysis of GW from ADM-Aeolus data: Banyard et al. (2021) presented a case study for GWs above the tip of South America using amongst others Aeolus data. There, the authors can take advantage of a very pronounced GW event with an amplitude in the range of 10 m/s, which is approximately twice as large as the averaged amplitude of the waves analysed in this case study (compare e.g. Figure 7). As the accuracy of the Aeolus data is in the range of these averaged amplitudes, the Aeolus data are not per se suitable for individual GW case studies. Focusing on pronounced events as Banyard et al. (2021) did or handling this challenge statistically by covering a large area (see Figure 2) and collecting around 100 profiles per day as we do are two possibilities to make use of Aeolus data for GW analyses.

As already briefly mentioned at the beginning of section 5, the mean residuals of the Aeolus measurements (see Figure 7) are in the same order of magnitude as those reported for other measurements in the literature (e.g., Figure 5 of Kramer et al. (2015), Figure 4 and 6 of Nath et al. (2009), or Figure 5 of Moffat-Griffin et al. (2017)). However, it must be mentioned here that those literature values are mostly residuals of individual measurements or amplitudes of GWs from radiosonde-based measurements and not averaged residuals. So, there might be a difference in the range of a small factor larger than 1. In comparison with literature values for the density of kinetic energy, the values derived here appear somewhat high (compare Figure 9 of Murphy et al. (2014) who mainly focus on vertical wavelengths of 2 – 3 km and shorter, for example or Figure 7 and 8 of Moffat-Griffin et al. (2017) who focus on vertical wavelengths of 13 km and shorter). Despite the same order of magnitude of the residuals this is possible because quadratic residuals are used to calculate the kinetic energy density. Thus, small differences will be amplified non-linearly. However, there are some radiosonde stations that show the same order of magnitude for the density of kinetic energy ($10^1$ J/kg) in their averaged values (see figure 4 and 5 of Yoshiki and Sato (2000)). Nevertheless, a conclusive comparison is difficult here. Firstly, the comparative measurements (radiosondes) are based on a different technique, i.e. they may address different volumes of air than Aeolus (miss-integration error). In addition, there is the movement of the measuring devices during the measurement and the different retrieval of the data. This can lead to a different



sensitivity with regard to GWs. Aeolus measures a horizontal line-of-sight velocity; this means that it is particularly sensitive to GWs whose air parcels oscillate horizontally (inertia GW) and parallel to the line of sight. Finally, the measurements were taken at different times and in different locations.

We find that $E_{kin,low}$ shows local maxima while a streamer event was present in the investigated geographical region. Nevertheless, $E_{kin,low}$ is not enhanced during the entire time (8. – 11. February 2020 and 3. – 7. November 2020). This can be due to the position of the specific wave sources or to the direction of wave propagation, which can be oblique, for example. To investigate whether an enhanced spatial resolution, at the cost of temporal resolution, could restrict the potential sources, the region is gridded with a resolution of 7.5° in latitude and 10° in longitude (see Figure 10). This corresponds to roughly

800 km in each direction for the mid-point of the addressed geographical region. If a temporal resolution of one week is chosen, the values per pixel range between 2 and 76 (14 – 24 values on average per pixel varying with the calendar week (CW)). Lower values would not be reasonable. The results for streamer 1, which shows a longer-lasting maximum in $E_{kin,low}$ than streamer 2, are depicted in Figure 10. Only during CW 43 – 45, nearly the whole geographical range (25 – 70° N, 45°W – 20° E) is covered. Especially, CW 47 and 48 suffer from larger data gaps. This is in accordance with Figure 8(c). The streamer event 1

lasted only some days and occurred in CW 45. Based on visual inspection, this week appears to be unremarkable in terms of possible local maxima of $E_{kin,low}$ or a generally enhanced averaged value over the whole plot. From Figure 8(c), which shows that $E_{kin,low}$ is relatively low after the maximum at Nov., $3^{rd} – 5^{th}$, an enhanced averaged value would not have been expected for CW 45. So, it can be noted that based on ADM-Aeolus data, local analyses do not reveal enhanced $E_{kin,low}$ during the time period of the streamer event 1. As the maximum of of $E_{kin,low}$ during streamer 2 is even shorter than during streamer 1, we

will not analyse it here.

Comparing Figure 8c) and Figure 9b) it is noticeable that the structure of Figure 9b) looks more regularly: maxima (of different height) can be observed every 3 – 4 days. In Figure 8c), the time interval between two succeeding maxima varies, one dominating oscillation cannot be identified. As mentioned earlier, streamers are due to enhanced planetary waves and streamers

are not the only GW source. The dominance of the 3 – 4 days signature in the time series of $E_{kin,low}$ during February 2020 suggests the conclusion that maxima of $E_{kin,low}$ are due to pronounced travelling planetary waves with a period of 3 – 4 days and that other GW sources are less important during that time period. According to Forbes et al. (1995), 3 – 4 days are typical for a travelling planetary wave with zonal wave number 2. So, Figure 9b) might show a regular effect of a planetary waves 2 on the generation of GW and the identified streamer event in February 2020 might be due to the most pronounced activity of

the planetary wave.

Here, it should also be stressed that only a part of the GW spectrum can be observed by ADM-Aeolus, as is the case for all instruments (e.g., Preusse et al. 2002; Wüst et al., 2006). Aeolus as a limb viewer looks to the side of ADM and collects all





information along the line of sight that is in our case mainly in zonal direction. GWs with phase fronts oriented meridionally will cancel out entirely or at least to a large part. Along the track, Aeolus averages approximately over 86 km (see section 3),

i.e., only GWs with (in our case) meridional wavelengths larger than 172 km can be detected in Aeolus data (Nyquist criterion). The larger the zonal wavelength and the smaller the meridional one, the closer the measured amplitude comes to the true amplitude. The effect of the vertical averaging is probably less important here. In conclusion, that underpins the role of $E_{kin,low}$ as a lower limit.

Finally, we come back to Figure 7 which depicts the averaged absolute residuals over height. As this figure shows a maximum at ca. 9 km height, which agrees approximately with the typical height of the wind maximum, we excluded the height range of 7 – 11 km from further analyses (as described in section 5) in order to be sure that we do not investigate any possible artefacts from a possibly insufficient detrending. However, this maximum at 9 km height does not need to be an artefact, it could also have a physical explanation. GWs generated at lower altitudes increase in amplitude as they propagate to higher altitudes. A

horizontal wind that also increases with height will filter the upward propagating GWs. Compared with lower altitudes, the greatest proportion of the upward propagating GWs will be filtered out at the height of the wind maximum. This means that it is possible that an increase in the average GW amplitude will be followed by a minimum. The same applies to the average residuals as extracted from Aeolus. At higher altitudes, the generation of secondary GWs, the generation of GWs due to strong wind shear at the height of the wind maximum or simply the decrease in air density can lead to an increase in the average GW

amplitudes and therefore in the Aeolus residuals.

Figure 11a) and b) depict the height of the first and the first as well as the second maximum per Aeolus wind profile. Both plots show lower numbers below 7 km height and strongly enhanced values above. Critical layer filtering will therefore mainly affect the height range above 7 km, below 7 km GW can grow in amplitude. The latter can be observed in Figure 7. The global maximum of the Aeolus wind profiles can indeed be found mainly at 8 – 9 km height, but also to a substantial part at higher

altitudes. Arguing that the wind maximum is on average at 8 – 9 km and that this leads to the maximum of the residuals at this height due to possible insufficient detrending, might therefore be too conservative. Including also the second highest maximum does not change the overall view tremendously.






**Figure 10 Development of the spatially resolved lower bound of the kinetic energy density averaged for the height range above 11 km with calendar week (CW), a) CW 43: October, 19th – 25th, b) CW 44: Oct. 26th – Nov. 1st, c) CW 45: Nov., 2nd – 8th, d) CW 46: Nov., 9th – 15th, e) CW 47: Nov., 16th – 22nd, f) CW 48: Nov., 23rd – 29th. The spatial resolution is 10° in zonal direction and 7.5° in meridional direction.**




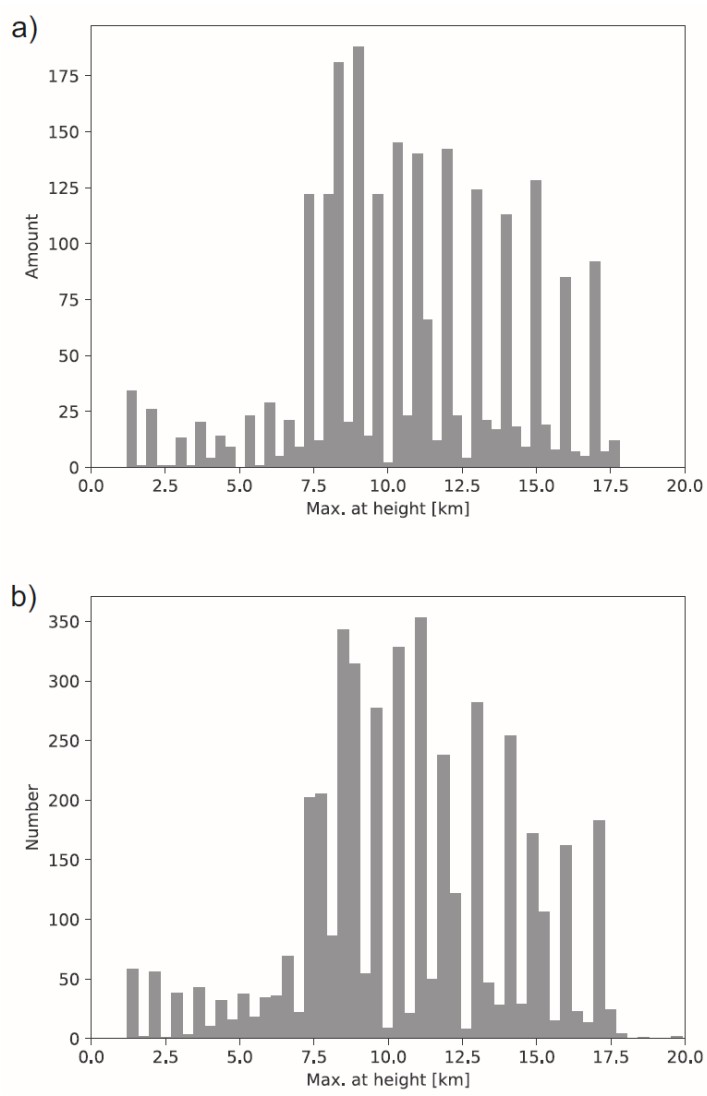

**Figure 11 While part a) shows the height of the maximum of each vertical wind profile used for the derivation of $E_{kin,low}$, part b) includes also the second highest maximum.**




## 7    Summary and outlook

The question, we addressed in this manuscript was, whether enhanced values of GW kinetic energy density (more precisely of

the lower bound for the kinetic energy density $E_{kin,low}$) over the Northern Atlantic and parts of Europe and Africa (25–70°N

and 45°W–20°E) can be observed during the passage of two streamers, one in February and the other one in November 2020,

based on ADM-Aeolus data.

Streamers, which are linked to several possible GW sources such as a pressure system, frontal activity and wind shear, are due

to enhanced planetary waves. They can be identified very well in TO3 maps from GOMOS or TROPOMI, for example. To

the best of our knowledge, they have not been analysed in the context of GW sources.

ADM-Aeolus measurements allow the derivation of $E_{kin,low}$ in the altitude regions where streamers are present. However, this

is challenging, as the GW-induced wind variations, which we derived by applying a repeating spline, lie in the same order of

magnitude as the accuracy of ADM-Aeolus. We tried to handle this challenge statistically by covering a large area and

collecting around 100 profiles per day. Comparison of this daily averaged $E_{kin,low}$ with the literature show that those values

are at the upper border. This emphasises the need for validation campaigns to take place at the same time and in the same place

and underlines the importance of specifying the systematic and the random part of an error.

In both streamer cases, we found significant local maxima in the daily resolved time series of $E_{kin,low}$ in the lower stratosphere

when the streamer passed the addressed geographical region. We also did a spatial analysis with a relative coarse spatial

resolution of 7.5° in latitude and 10° in longitude (roughly 800 km in each direction for the mid-point of the addressed area).

In order to get a reasonable amount of data per pixel, we applied a temporal resolution of one week. However, this resolution

is not good enough to get more precise information about the region of enhanced $E_{kin,low}$ and therefore about the possible GW

generation mechanism.

In this manuscript, we focused only on the kinetic energy, however, GW transport not only kinetic energy, they also carry

potential energy. For low-frequency waves, the relation of the horizontal kinetic energy densities and the potential energy

density, $E_{kin,h}$ and $E_{pot}$, is determined by

$$\frac{\overline{E_{kin,h}}}{\overline{E_{pot}}} = \frac{1 + \left(\frac{f}{\hat{\omega}}\right)^2}{1 - \left(\frac{f}{\hat{\omega}}\right)^2}$$

(4)

(Geller and Gong, 2010). In their appendix B, Geller and Gong (2010) provide the above mentioned relation using the full GW

dispersion relation. The authors state that the asymptotic behaviour of the equation above based on the full dispersion relation

and the dispersion relation for low-frequency waves is similar. Since a succeeding mission for ADM-Aeolus is planned, which

reaches also higher altitudes, information about $E_{kin,h}$ might be available even higher up in future. For the calculation of $E_{pot}$,

vertical temperature profiles are needed. Those are currently available through TIMED-SABER or GPS radio occultation





satellites. Given that the wind and the temperature measurements address approximately the same air volume at the same time, which is not the case for TIMED-SABER and ADM-Aeolus in the vast majority of cases, $\hat{\omega}$ can be estimated. $\hat{\omega}$ is an important

parameter for calculating further GW information such as ratio of the vertical to the horizontal group velocity for low-frequency waves or the angle between lines of constant phase and the vertical for high-frequency waves, etc. In conclusion, it would be helpful for GW analyses, if ADM-Aeolus is synchronized with a temperature mission.





**Author contribution**

This work was funded by two projects: LISA and WAVE. The first one was acquired by LK, SW and MB, the second one by SW and MB. The streamer events were identified by FT and LK and checked by SW. The GW analyses were performed by SW. The manuscript was written by SW. All authors read the manuscript.

**Acknowledgement**

The work of Sabine Wüst was funded by the European Space Agency, ESA, (project LISA, ESA Contract No. 4000133567/20/I-BG) and the Bavarian State Ministry for the Environment and Consumer Protection (WAVE, TKO01KPB-73893). The work of Lisa Küchelbacher was funded in parts by the European Space Agency, ESA, (project LISA). We thank Oleg Goussev and Isabell Krisch (both DLR) for their support concerning the Aeolus data download.

**Code / Data availability**

The detrending algorithm is described in Wüst et a. (2017). The ADM-Aeolus data was downloaded from the ESA Aeolus
Online Dissemination System (https://aeolus-ds.eo.esa.int/oads/access/, last download: 1$^{st}$ Dec. 2020).

**Competing interest**

The authors declare that they have no conflict of interest.





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
