# Peer review of "Gravity waves above the Northern Atlantic and Europe during streamer events using Aeolus"

_Atmospheric Measurement Techniques, 2024_

## Author Comment (AC2)

*Answer to the comments of referee 3 (RC 1):*

We would like to thank the anonymous reviewer for his / her valuable comments and additional information. I found them very helpful and changed the manuscript accordingly in most cases.

Two further remarks: when I read the manuscript, I found three further minor issues, which I corrected:

- In table 1, I changed the unit of the variable "alt" to from "M" to "m". Furthermore, I replaced "RD7" in row 4 of this table by "ADM-Aeolus Level-2B Algorithm Theoretical Baseline Document"
- p.14, I. 295: I changed "section 5.1" to "section 5". This was a relict of a previous version.

I tried to avoid page and line numbers but where I used them I refer to the new version with accepted changes.

**RC1**: 'Comment on amt-2024-18', Anonymous Referee #3, 10 Apr 2024  reply

In their paper "Gravity waves above the Northern Atlantic and Europe during streamer events using ADM-Aeolus" the authors use ADM Aeolus data to derive the kinetic energy density (Ekin) of gravity waves over the Northern Atlantic and Europe during periods of streamer events. During these events enhanced gravity wave energy densities would be expected. One major problem is the high noise level of Aeolus which complicates the derivation of gravity wave signals. Therefore, the authors average over a larger region to reduce the uncertainty of time series of daily averages for relating them to the streamer occurrences. Indeed, minor enhancements of Ekin are found that may be related to the streamers. Horizontal distributions of Ekin show enhancements that may be related also to other gravity wave sources.

Overall, the paper is an interesting study, fits into the scope of AMT, and the results are of interest for the community of atmospheric dynamics and people who intend to derive gravity wave distributions from Aeolus soundings.

The paper is therefore recommended for publication in AMT after addressing my minor, but important comments.

**MAIN COMMENTS** are:

(1) The expression "Ekin,low" is somewhat misleading and should be avoided! This expression as is suggests to be a lower estimate of the gravity wave kinetic energy density. However, Aeolus Ekin estimates should be significantly high-biased because of the high Aeolus noise level.

Therefore it should be stated more clearly in the manuscript that Ekin derived from Aeolus should not necessarily be a lower estimate!

Using a better statistics will reduce the "uncertainty" of Ekin, but will not remove the bias introduced by Aeolus noise. Of course, a better statistics will help to obtain more reliable relative variations of Ekin assuming that the noise produces a constant Ekin offset.

Removed expression Ekin,low from whole manuscript including figures, re-formulated abstract

(2) Gravity waves found in the region of interest are not necessarily excited in the same region by the streamer events. It has been shown before by, for example Krisch et al. (2017) that gravity waves can travel large distances horizontally until where they are observed. This information should be added in the discussion.

Done

**SPECIFIC COMMENTS:**

(1) l.15 and l.21: Later in the manuscript it turns out that Ekin derived from Aeolus winds should not necessarily be a lower limit. Therefore these statements in the abstract should be revised.

Done

(2) l.32: please add the citation Holton, 1982 in addition to Houghton, 2002

Holton, J. R.: The role of gravity wave induced drag and diffusion in the momentum budget of the mesosphere, J. Atmos. Sci., 39, 791-799, 1982.

Done

(3) l.32/33, l.41 onward: Please be more specific! For a gravity wave pseudomomentum flux (Fpx,Fpy) is conserved, but not the wave energy in general. Specifically, in your paper you are considering Ekin, which is not conserved! Even if a wave propagates conservatively, Doppler-shifting by the background wind will change the intrinsic frequency, which then leads to changes in wave kinetic energy.

Sorry, here I don't get your point. In the first paragraph of the introduction I mention that the pseudo-momentum and the pseudo-energy are the conserved quantities, if the background wind is unequal to zero. Isn't that what you wanted to be mentioned? So, could you please specify your comment? For consistency reasons, I included the info how to derive the pseudo-energy from the energy density. Now, there are the formulas for the pseudo-energy and the pseudo-momentum mentioned in the manuscript.

(4) l.51: momentum -> pseudomomentum

Done, also replaced it in the line following the equation.

(5) l.84: accuracy -> precision (see comment (8))

In principle, you are right, challenging is the precision. However, ESA only provides the error, therefore, I replaced accuracy with "error especially the precision"

(6) l.85: This measurement noise will result in a high-bias of Ekin.

Not necessarily. If the noise is due a bias constant with height, the detrending method will remove most of it. If the noise is due a lack of accuracy, we will get a higher error in E_kin. I changed the sentence to "One challenge is the accuracy error (especially the precision, since it is not removed through the detrending procedure in contrast to a bias) of Aeolus, which is lower than originally planned for and now in the same order of magnitude as typical GW fluctuations."

(7) l.127-130: To my knowledge, Metop-A is no longer operational. You should refer to the GOME-2 instruments on Metop-B and Metop-C, instead!

Metop-A was operational until 2021 (see e.g. right column, upper part of https://space.oscar.wmo.int/satellites/view/metop_a), the decommission started on Nov., 15th 2021 (https://www.eumetsat.int/europes-first-meteorological-satellite-polar-orbit-endsits-run). For our analysis, we used a combination of Metop-A and -B data in order to reduce the number and size of data gaps. We changed the sentence (p. 5, l. 131) "MetOp-AB was launched in 2006" to Metop-A was launched in 2006 (Metop-B in 2012)" to make the info more precise.

(8) l.223: What do you mean by "accuracy"? Usually, in the field of science and engineering "accuracy" refers to systematic errors/biases, while "precision" refers to random errors. See also: https://en.wikipedia.org/wiki/Accuracy_and_precision Alternatively, you could use the notation after ISO 5725 (which is rarely used in our field), but this should then be stated clearly and, accordingly, the expression "bias" should be generally avoided. Please check the manuscript throughout for the correct use of "accuracy" and "precision"!

Sorry, that was confusing. ESA provides the error and does not specify whether it is due to bias or precision. I went through the manuscript and revised it accordingly.

(9) L.264/265: This assumption is not valid for polar latitudes. Please refer to Krisch et al., AMT, 2022 for details.

Krisch, I., Hindley, N. P., Reitebuch, O., and Wright, C. J.: On the derivation of zonal and meridional wind components from Aeolus horizontal line-of-sight wind, Atmos. Meas. Tech., 15, 3465-3479, https://doi.org/10.5194/amt-15-3465-2022, 2022.

Yes, that's true and the reason why I wrote "to a large extent". I inserted "(this does not hold for polar latitudes, see Krisch et al. (2022))" to make it clearer.

(10) l.265: The expression Ekin,low is misleading as Ekin is calculated from squared wind fluctuations such that random noise will not average out and will produce biases. This should be particularly the case for Aeolus.

Ok, I get your point. I deleted all occurrences of E_kin,low and replaced them by E_kin.

(11) l.293 onward: please check for the use of "accuracy" and "precision", see comment (8)
Done

(12) Fig.6: "accuracy" should be "precision", or just "error"??
It should be precision I changed it.

(13) l.333-336: I think it is encouraging that the positive and negative mean residuals are about symmetric with respect to zero, suggesting that incomplete removal of the background does not introduce strong biases on average. This could be mentioned in the manuscript.

Thanks for the hint. I inserted a sentence in the description of figure 7 in the manuscript. Additionally, I clarified at the beginning of the description of figure 7 that the first part refers to figure 7a.

(14) l.401-412: In addition, Aeolus noise will produce a considerable offset of Ekin causing a high-bias.

That's true. I inserted "They could be the results of the relatively high Aeolus error, specifically the if it is due to a low precision." in line 412.

(15) l.415-430: It is remarkable that in Fig.10 enhanced Ekin is found in the vicinity of Greenland. The southern part of Greenland is a prominent source of gravity waves as can be seen in different gravity wave climatologies (for example, Hoffmann et al., 2013; Ern et al., 2018).

Hoffmann, L., Xue, X., and Alexander, M. J.: A global view of stratospheric gravity wave hotspots located with Atmospheric Infrared Sounder observations, J. Geophys. Res. Atmos., 118, 416-434, doi:10.1029/2012JD018658, 2013.

Ern, M., Trinh, Q. T., Preusse, P., Gille, J. C., Mlynczak, M. G., Russell III, J. M., and Riese, M.: GRACILE: a comprehensive climatology of atmospheric gravity wave parameters based on satellite limb soundings, Earth Syst. Sci. Data, 10, 857-892, https://doi.org/10.5194/essd-10-857-2018, 2018.

After pointing out that the streamer cannot be seen in figure 10, I included "However, in three out of six weeks enhanced E_kin values are observed near Greenland. This area is known to be a prominent GW hotspot (Ern et al., 2018; Hoffmann et al., 2013)."

(16) l.415-430: You should mention in the discussion that gravity waves can travel larger distances from their source to the location where they are observed (for example, Krisch et al., 2017). Therefore gravity waves of origin other than the streamer events could be superimposed in the region of interest and partly obscure the streamer gravity wave signal. Possible candidates for gravity waves of different origin could be mountain waves from Greenland and Iceland.

Krisch, I., Preusse, P., Ungermann, J., Doernbrack, A., Eckermann, S. D., Ern, M., Friedl-Vallon, F., Kaufmann, M., Oelhaf, H., Rapp, M., Strube, C., and Riese, M.: First tomographic observations of gravity waves by the infrared limb imager GLORIA, Atmos. Chem. Phys., 17, 14937-14953, https://doi.org/10.5194/acp-17-14937-2017, 2017.

Following the previous change, I added the following "Since GW can travel large distances from their source (e.g., Krisch et al., 2017), it cannot be excluded that orographic GWs generated through airflow over Greenland contribute to the time series of E_kin."

(17) l.442: This is not correct! Aeolus is not a limb viewer! It views downward with an angle of just 35deg to the nadir.

I corrected it.

(18) l.443/444: "GWs with phase fronts oriented meridionally will cancel out entirely or at least to a large part." This is not correct because Aeolus observes in "near-nadir" geometry!

I changed it to "Aeolus as an off-nadir viewer (35° incidence angle) looks obliquely through the atmosphere and collects all information along the line of sight. GWs with phase fronts not oriented parallel to the line of sight are displayed attenuated in the Aeolus data". Furthermore, I deleted "The effect of the vertical averaging is probably less important here".

(19) l.456: However, gravity waves of phase speed opposite to the background wind would not be filtered out. These gravity waves could undergo considerable amplitude growth and overcompensate the filtering effect.

Yes, I agree. However, as the text is written, this possibility is not excluded. This is to stress that there may also be a physical explanation for this observed maximum in average residuals plotted against height.

(20) l.458: "the generation of secondary GWs" This statement is very speculative. Of course, this process can also happen at relatively low altitudes, however, usually it becomes important at altitudes well above 20km not covered by Aeolus.

Yes, this is a speculative statement. I changed the order, so mentioned the decrease in air density first and the secondary GWs afterwards. Additionally, I used "could" instead of "can" to emphasize the speculative character.

(21) l.495: The relatively high values of Ekin are understandable because the high noise level of Aeolus will cause an offset.

I inserted „This can be the result of a low precision of the Aeolus measurements."

(22) l.513: There are more satellite instruments that currently observe temperature altitude

profiles in the lower stratosphere and are suited for gravity wave analysis, for example, AIRS and MLS, see Hoffmann and Alexander (2009) and Ern et al. (2022).

Hoffmann, L., and Alexander, M. J.: Retrieval of stratospheric temperatures from Atmospheric Infrared Sounder radiance measurements for gravity wave studies, J. Geophys. Res., 114, D07105, doi:10.1029/2008JD011241, 2009.

Ern, M., Hoffmann, L., Rhode, S., and Preusse, P.: The mesoscale gravity wave response to the 2022 Tonga volcanic eruption: AIRS and MLS satellite observations and source backtracing, Geophys. Res. Lett., 49, e2022GL098626, https://doi.org/10.1029/2022GL098626, 2022.

I get your point, it looks like this should be a complete list but that wasn't my intention. I inserted "for example".

**TECHNICAL COMMENTS:**

(1) l.23 due to -> caused by
Corrected

(2) l.25: between the daily averaged -> between enhanced daily averaged
Corrected

(3) l.33 are can -> can
Corrected

(4) l.59: by -> be
Corrected

(5) l.71: doppler -> Doppler
Corrected

(6) l.81: data has -> data have
Corrected

(7) l.112: air of low -> air of low total column ozone (TO3) ???
Meant is "which transport air from low latitudes into mid latitudes", I changed it accordingly.

(8) l.121: The identification -> Our identification
Corrected

(9) l.134: comparable large -> comparably large
Corrected

(10) l.139: longitudes, so it has a strong meridional structure -> latitudes, so it has a strong zonal structure

[Figure]

Are you sure? I mean this streamer:

(11) caption of Fig.1: starting data -> starting date
Corrected

(12) l.147:  November 2020 ->  September 2020
Corrected

(13) l.180: horizontal position -> horizontal orientation
Corrected

(14) Table 1: dependant  -> dependent
Corrected

(15) l.397: ofKramer -> of Kramer
Corrected

(16) l.438 waves 2 -> wave 2
Corrected

---

## Author Comment (AC3)

*Answer to the comments of referee 2 (RC 2):*

We would like to thank the anonymous reviewer for his / her valuable comments and additional information. I found them very helpful and changed the manuscript accordingly in most cases.

Since the other reviewer had some serious concerns about the nomenclature "E_kin,low" (Aeolus doesn't have the best precision, which might lead to a bias in the derivation of E_kin; that's why "E_kin,low" might be misleading"), I changed it to "E_kin". This is just an info in order to reduce the risk of confusion when having the first version of the manuscript in mind.

Two further remarks: when I read the manuscript, I found three further minor issues, which I corrected:

- In table 1, I changed the unit of the variable "alt" to from "M" to "m". Furthermore, I replaced "RD7" in row 4 of this table by "ADM-Aeolus Level-2B Algorithm Theoretical Baseline Document"
- p.14, l. 295: I changed "section 5.1" to "section 5". This was a relict of a previous version.

I tried to avoid page and line numbers but where I used them I refer to the new version with accepted changes.

Throughout this review, the format "PXX.LXX Comment" is used to refer to the page number "PXX" and line number "LXX" in the originally submitted manuscript corresponding to each comment.

**SHORT SUMMARY**

This is a review of the paper titled "Gravity waves above the Northern Atlantic and Europe during streamer events using ADM-Aeolus", submitted to Atmospheric Measurement Techniques. This paper investigates the potential for Aeolus measurements to provide gravity wave (GW) signals during so-called "streamer events" above the Northern Atlantic and Europe, and in particular, to derive a lower limit for the GW kinetic energy density (E_kin_low) during such events. Two example cases are analysed, and a temporal correlation is found between the daily averaged E_kin_low and the occurence of each streamer event, with enhanced values observed for each. The authors also consider the spatial distribution of the kinetic energy density signals from Aeolus during one of these events, however no significant pattern can be found. Cubic splines are used to approximate the atmospheric background and retrieve wind residuals, and the analysis is split into a tropospheric and a stratospheric part to exclude the tropopause wind maximum.

This manuscript is interesting and has the potential to fulfil the scope of AMT, the results are presented in a balanced manner and some of the scientific quality is good. However, a major revision is likely to be necessary in order to address the following three issues.

(i) The study itself is rather focused and could be improved by both a broader development of the GW analysis technique and a wider analysis of the streamer events mentioned and/or of other similar events.

(ii) There are some outstanding questions regarding the validity of the analysis technique for measuring GWs, with the inherent uncertainties in the Aeolus data. This may just require some clarification.

(iii) Although there is a good clarity and concision to the overall writing of the manuscript, the figures and presentation of results will require further improvement to a higher quality.

Therefore, publication can be recommended only after the following issues and suggestions are resolved or considered.

**GENERAL COMMENTS**

1) The term "ADM" (abbreviation of Atmospheric Dynamics Mission) was omitted from the satellite's name by ESA several years ago. Please update all instances of "ADM-Aeolus" to "Aeolus" to conform with convention, noting this previous AMT review in particular, general comment #2: https://egusphere.copernicus.org/preprints/2023/egusphere-2023-1924/egusphere-2023-1924-RC1-supplement.pdf

Corrected

2) P01.L23 With respect to issues (i) and (ii), it is not completely obvious how sharp regions of vertical wind shear, which may define the slanted edges of jet streaks and other non-GW tropospheric wind phenomena, are prevented from contaminating the signals appearing in the GW analysis of Aeolus data. Are these non-GW features captured correctly in the background using the cubic spline method? This is particularly an issue in the troposphere, since the stratosphere is generally stable and stratified, such that vertical perturbations in horizontal wind can be more easily attributed to GWs here. Attached to this review is an along-track Aeolus "quick-look" profile time-series for 2020-02-10 in the region of interest, showing an intense jet streak associated with one of the streamer events in question (Source: https://aeolus.services/). Might there be an increase in non-GW wind perturbations, which inherently occurs during these dynamic streamer events, which could contribute to the results and/or explain some of the temporal coincidence that is found? If this is accounted for, some clarification of this within the text would be helpful. Optional Suggestion: Perhaps an example of the raw Aeolus data from the time period being analysed could be shown at some point to help orient the reader?

When extracting gravity waves from vertical profiles (this includes not only wind, but also other parameters such as temperature or ozone), it is unfortunately a common problem: not every signal in the residuals can necessarily be attributed to a gravity wave. Individual residual profiles can be checked by eye for unusual signals, but for a large number of profiles, some thousands as is the case here, this is only possible in parts. This is also what I did: I looked at some de-trended Aeolus profiles. In many cases, the peak in the tropopause fits well into the oscillation observed before and after, i.e. it does not "disturb" the vertical wavelength (see attached figure, this is an Aeolus profile from 4th Nov. 2020 (time period of streamer 1) on the left side and the detrended version on the right side).

Furthermore, in order to reduce the risk of signals in the residuals that are not attributable to gravity waves, it is helpful to exclude typical areas that lead to the generation of "artificial" signals in the residuals. The most important areas here are the atmospheric pauses, in our case the tropopause. That is why I restricted the analysis to the height range below 7 km and above 11 km.

Additionally, I used a method that can better reproduce strong deflections than the classical spline method due to the variation of its starting points.

In order to make this whole topic a little bit clearer, I included an additional part in the discussion where I specifically address this problem (p. 23, ll. 474 ff.).

[Figure]

3) P23.L450 - P23.L460 The question of whether or not the detrending method is affected by the tropopause wind maximum as an artefact, contaminating the GW signal, is quite an important one. Do the authors think this peak in residual winds is a consequence of this artefact, or is it more attributable to GW filtering, as this paragraph suggests? Will Aeolus have the capability to measure an increase in GW amplitudes which is caused by secondary GWs? An expansion of this discussion and perhaps an exploration of further GW statistics would be useful, particularly to answer these and similar questions.

We cannot say with certainty whether the peak in the area of the tropopause is due to gravity waves or to insufficient de-trending in the area of the tropopause. For this reason, I excluded this area from the analyses as described at the beginning of section 5, but also pointed out in the discussion (section 6, the part you address here) that there may well be physical reasons for this peak. A hint that the detrending performs not too bad is that both, negative and positive residuals versus height (Figure 7b) are highly symmetrical: this suggests that, at least on average, the detrending does not introduce strong biases. This is now mentioned at the beginning of section 5 (p. 16, ll. 326 ff). Whether Aeolus is sensitive to secondary waves depends on the amplitude of those waves.

An expansion of the discussion would be of interest, however, from my point of view, it would be highly speculative.

4) P26.L484 The question of the manuscript seems rather focussed to me, and could be expanded and generalised a little so that this work is more applicable as an atmospheric measurement technique for future gravity wave analyses using Aeolus. Whether this is formed from a slightly broader GW analysis, looking deeper than the kinetic energy density, or whether additional events are required to test the conclusions that have been drawn; both may help to answer some of the current questions that remain about this study.

The question in this manuscript is indeed focused. We focused on one possible source of gravity waves (streamers, which to our knowledge have not been explicitly addressed as a source of gravity waves in the literature) and selected the two most clearly observable events that fell within the project period. Streamers are not known to be a strong source of gravity waves, and there is only one publication dealing with gravity waves from Aeolus. This is an extremely high amplitude gravity wave. Due to the spatial extent of a streamer event and the challenging data quality of Aeolus for these analyses, it was necessary to examine a relatively large spatial area, and even in this case, i.e. when analysing the two clearest streamer events, only a significant but not unusual increase in the kinetic energy density in the stratosphere is seen. The combination of TIMED-SABER, e.g. to derive the potential energy density, adds little or no value due to the different overflight times and locations. The extension to other gravity wave sources would dilute the focus of the manuscript.

5) Are the authors aware of work such as Wiegand and Knippertz, 2013 (https://doi.org/10.1002/qj.2112), Wernli and Springer, 2007 (https://doi.org/10.1175/JAS3912.1) and Madonna et al., 2014 (https://doi.org/10.1175/JAS-D-14-0119.1)? These may be useful to read and as a references.

Thank you for the citations. Madonna et al. (2014) refer to potential vorticity (PV) streamers as defined by Appenzeller and Davies (1992), see first page second column of their publication. We refer to another term of streamers following Offermann et al. (1999). Krüger et al. (2005), who investigate the same kind of streamers as Offermann and we, write in their manuscript: "The term used in this study should not be mistaken with the terminology of "streamers" of a smaller-scale first introduced by Appenzeller and Davies (1992) and Appenzeller et al. (1996), describing stratospheric intrusions into the troposphere." Also, Wernli and Sprenger (2007) and Wiegand and Knipperts (2013) refer to PV vorticity streamers.

**SPECIFIC COMMENTS**

P01.L16 Replace "Aeolus on ADM" with "Aeolus". ALADIN is the instrument onboard, as mentioned later in the article.

Done

P01.L17 (also P03.L77) The phrase "from the ground to the lower stratosphere (20 - 30 km)" is a little ambiguous. The ground is of course at 0 km, and the stratosphere also begins lower than 20 km, so this should be rephrased slightly.

I changed it to „ from the ground to an altitude of ca. 20 – 30 km ..."

P01.L25 Do the results indicate this, or is this an implicit assumption? As the authors mention later, Banyard et al. 2021 has shown that this is possible, although only for the single case of a strong GW.

That's a good point. I changed the sentence to "The derivation of GW signals based on Aeolus data is possible, however, we collected about 100 profiles to statistically reduce the uncertainty of the daily averaged E_kin."

P02.L31 It's slightly confusing to say that GWs dominate atmospheric dynamics over other wave phenomena, even if this is qualified by specifying that this is "especially above 75 km height". Perhaps this part could be rephrased? This study does not go higher than 24 km.

I changed the sentence to "Especially above 75 km height, GWs dominate atmospheric dynamics through the deposition of energy and momentum, even though there exist wave phenomena with larger periods and wavelengths in the atmosphere (Houghton, 2002)."

P03.L84 The phrase "Challenging is the accuracy of Aeolus" reads a little strangely.

I changed it to "One challenge is the accuracy of Aeolus,…"

P05.L107 Can streamer events be linked to strong cyclones?

Good question. In our cases, the dominating pressure system was an anticyclone (see also https://earth.nullschool.net/#2020/11/04/0900Z/wind/isobaric/250hPa/orthographic=-20.72,38.04,312 for 4[th] Nov. 2020, 250 hPa, and

https://earth.nullschool.net/#2020/02/09/0900Z/wind/isobaric/250hPa/orthographic=-20.72,38.04,312 for 9[th] Feb. 2020, 250 hPa).

In this manuscript, we focus on tropical-subtropical streamers, i.e. air is transported northwards from the tropics. As can be seen from the two internet addresses given, the zonal jet is strongly deflected northwards here. Accordingly, there should be an anticyclone to the east of the zonal jet. I would therefore expect this type of streamer to be identified with high pressure areas. However, as I have not found any relevant literature on this and as we only treat one specific kind of streamer (so not the polar ones) in this manuscript, I changed "anticyclone" to "pressure system" in the text to keep it more general.

P05.L112 low latitude or low potential vorticity?

Clarified (low latitudes).

P08.L172 Could you please clarify how you corrected the hlos wind for the satellite observation geometry?

We already explained it in the following section. I added a hint towards it in brackets.

P08.L176 - P08.L185 Could this paragraph be written a little more succinctly? It's easy to get lost here at the moment.

I rearranged this paragraph – it is not shorter now but I hope that it is easier to follow the argumentation.

P12.L251 For clarification, is the use of varying starting points applied in order to combat both the insufficient approximation of extrema in the background and the artificial oscillations generated?

Yes, it is.

P13.L268 This sentence is a little unclear, could it be rephrased slightly?

I changed it to "Based on this algorithm, maxima of mean $E_{kin}$ over the area or time period defined at the beginning of the algorithm can be identified, thus GW hotspots in place or time."

P14.L291 By "CIRA" are you referring to the COSPAR International Reference Atmosphere? This needs defining.

Yes and done.

P21.L378 Is it possible to suggest reasons for the GWs captured in the analysis of these two streamer events? E.g. Is it unstable shears, geostropic adjustment, the result of interaction between other GWs? It would be good to have some additional discussion on this topic here.

That's a little bit tricky, since it is more speculation than knowledge, as we have no information about the precise location of the GW we observe and therefore can't trace them back. I added the following: "GFS data at 250 hPa, for example, show a strong anticyclone, which is linked to downward vertical movement in the centre near the position of the streamer in both cases investigated here (e.g. https://earth.nullschool.net/#2020/11/04/0600Z/wind/isobaric/250hPa/orthographic=-29.56,45.15,312 for streamer 1, and https://earth.nullschool.net/#2020/02/09/0600Z/wind/isobaric/250hPa/orthographic=-29.56,45.15,312 for streamer 2, last access: 27th May 2024). Additionally, a strong vertical shear of the horizontal wind can be observed when addressing different heights. Also frontal activity is present (https://www.wetterzentrale.de/reanalysis.php?jaar=2020&maand=11&dag=4&uur=000&var=45&map=1&model=dwd for streamer 1, and https://www.wetterzentrale.de/reanalysis.php?jaar=2020&maand=2&dag=9&uur=000&var=45&map=1&model=dwd for streamer 2, last access: 27th May 2024)."

P22.L442 Aeolus measurements are "off-nadir", even though ALADIN looks to the side of Aeolus, it is not a limb viewer.

I changed it to "Aeolus as an off-nadir viewer (35° incidence angle) looks obliquely through the atmosphere and collects all information along the line of sight.".

P23.L447 It might be useful to explain here a little more about why it is the lower limit of the kinetic energy density that you are measuring.

I changed it to: "In conclusion, that underpins that E_kin is strictly speaking only a lower limit: this is due to the effects just mentioned and to the fact that the instrument measures only along the line of sight and therefore not the whole 3D wind vector. However, a low precision of the Aeolus measurements can lead to a significant bias in the derived E_kin values."

P26.L503 - P27.L517 This is a good and useful discussion which highlights some of the potential future benefits of a succeeding mission to Aeolus and raises important suggestions for optimising its suitability for GW analysis.

Thank you! I appreciate.

**FIGURE COMMENTS**

Figure 1 The latitude and longitude labels need to be larger and not obscured by the colour bar. Is there a higher resolution version of the image shown?

The latitude and longitude labels are now larger and not obscured by the colour bar any more. The resolution is improved.

Figure 2 Please add latitude and longitude labels.

Done. To make it more concise, we use the same projection as for figure 1 now.

Figure 7 It could be made a little clearer within the plot itself that 7b shows the negative and 7c the positive residuals.

Done

Figure 8 Since 8c is just the green plot from 8b repeated, is there a way to simplify the reading of the entire figure somehow? Having labels on the figure to quickly identify which subplots correspond to the tropospheric and stratospheric parts may be helpful.

Done

Figure 9 Can these figures be superimposed on one another, or is it better to separate the tropospheric and stratospheric components for all figures? It feels like this could be presented in a slightly clearer way, as you have to look quite carefully to understand what each subplot shows at the moment.

Combined in one figure

Figure 10 The filled contour plotting here is a little confusing, would a different colour scale be better to use? Perhaps a different plotting technique and layout would make it easier to see the changes from week to week? Please put the dates as well as the week numbers.

I inserted the dates. When I created this figure, I tried different colour scales. In the end, I chose this one because I think it is relatively intuitive (low values = blue, high values = red, another colour for medium values).

Figure 11 It is not immediately clear that these figures show the heights of the profile's first and second wind maxima. Is it possible to show this as some sort of density plot with height on the y axis and two variables on the x axis (Var 1 being the height of the first maxima, Var 2 being the heights of the first and second maxima)? Histograms may be fine if they are slightly altered, but they are a bit unclear at present.

I changed the histograms. For me, they seem to be more intuitive now.

All Figures: In general, most of the figures need some improvement in quality to make them clearer and easier to understand. An additional figure to give more context for Aeolus measurements may be helpful, and a couple more may be required for any further analysis that is conducted.

**TYPOGRAPHICAL ERRORS**

PXX.LXX New (Old)

P02.L33 which can (which are can)
Corrected

P02.L35 GWs (GW)
Corrected

P02.L37 GWs (GW)
Corrected

P02.L38 improving (improve)
Corrected

P03.L59 However, parts of the GW spectrum, equation (3) can be simplified (However, for parts of the GW spectrum equation (3) can by simplified)
Are you sure? I would like to express that equation (3) holds only for s specific part of the GW spectrum

P03.L63 frequently (frequent)
Corrected

P03.L71 Doppler (doppler)
Corrected

P03.L75 Aeolus (Aeolus on ADM)
Corrected

P03.L81 These data have (These data has)
Corrected

P05.L135 clearly (definitely)
Corrected

P06.L139 reaches latitudes of 70 degrees N (reaches regions latitude of 70 degrees N)
Corrected

P08.L174 netCDF (ncdf)
Corrected

Check Kruger
Does this comment really refer to this page and line? Is it maybe an artefact?

P12.L243 adaptation (adaption)
Corrected, also later in the manuscript (p. 14, l. 291)

P20.L365 Figure 7 (Figure 6)
Corrected

P26.L484 The question which we addressed in this manuscript was whether (The question, we addressed in this manuscript was, whether)

Corrected

---

## Referee Report (RR1)

**SHORT SUMMARY**

The paper titled "Gravity waves above the Northern Atlantic and Europe during streamer events using Aeolus" has been revised somewhat to address points raised by this and other reviewers' comments. It has been improved, however there are still a number of issues that this reviewer has with the manuscript in its current form. These would need addressing before publication by AMT can be recommended by this reviewer.

**GENERAL COMMENTS**

1) This reviewer still has concerns about the method used to extract GW perturbations and there is no substantial development of the analysis technique or a wider analysis which might help to diminish these concerns. The authors have instead sought to clarify in the discussion the reasons for their previous method choice, which they do by considering the tropopause as the main non-GW perturbation source, and by using the 'repeating spline' method to avoid 'strong deflections'. Although the reviewer would prefer to see a consideration for other important wind perturbation sources in the troposphere, the authors do now clearly discuss this limitation of their method. Indeed, the authors have added some caveats into the text to clarify the shortfalls of the GW extraction method as a consequence of the techniques they are using.

2) The focus of the study has not been widened much in response to the previous comments by all reviewers. It seems a little unusual to focus on just streamer events (and one specific type at that – see author comment on P05.L107 on page 5 of their response to RC2) and only using Aeolus, for an AMT paper, which is why the suggestion to widen this focus was made previously. Would this article be better suited to a different journal if it is to be so focused, as the authors have acknowledged that it is?

3) Further changes are necessary to some of the figures before they are of publishable quality. Notably, figure 10 is very difficult to understand with the filled contour method currently used to plot the data. Although the authors have clarified the dates corresponding to each week and have defended their colour scale choice, they have not changed the plotting technique to make it easier to read. Furthermore, it is very difficult to follow the story being told by this figure, in tandem with the corresponding text.

**SPECIFIC COMMENTS**

PXX.LXX Comment

P03.L63 However, for parts of the GW spectrum, equation (3)... (Please insert a comma after 'spectrum' to avoid ambiguity.)

P03.L87 The phrase "especially the precision, since it is not removed through the detrending procedure in contrast to a bias" is now a little confusing. Why is the precision being removed? For Aeolus, which is more challenging, the precision or the accuracy?

P08.L178 This now reads a bit better, the authors have clarified that the variable being added is only being corrected for the satellite's orbital node and it is clearer.

**FINAL COMMENT**

Overall, no significant changes have been made to the scientific method, scientific quality or presentation quality for this manuscript, and so it is not yet possible to recommend that the article be accepted for publication without some of the changes outlined above and previously. There have been some good alterations to the text which add important clarifications, however further revisions to the method and quality are still required for this reviewer to accept the paper.

---

## Author Response (AR2)

**Report #1**

Submitted on 26 Jun 2024 Anonymous referee #3

Anonymous during peer-review: Yes No

Anonymous in acknowledgements of published article: Yes No

**Checklist for reviewers**

| <ol> <li>Scientific significance</li> <li>Does the manuscript represent a substantial contribution to scientific progress within the scope of this journal (substantial new concepts, ideas, methods, or data)?</li> </ol>                                                                           | Excellent Good Fair Poor |
|------------------------------------------------------------------------------------------------------------------------------------------------------------------------------------------------------------------------------------------------------------------------------------------------------|--------------------------|
| 2) Scientific quality
Are the scientific approaches and applied methods valid? Are the results discussed in an appropriate and balanced way (consideration of
related work, including appropriate references)? Note that papers do not necessarily need to be long to be scientifically sound. | Excellent Good Fair Poor |
| 3) Presentation quality
Are the scientific results and conclusions presented in a clear, concise, and well structured way (number and quality of figures/tables,
appropriate use of English language)?                                                                                         | Excellent Good Fair Poor |

**For final publication, the manuscript should be**

accepted as is

accepted subject to **technical corrections** accepted subject to minor revisions reconsidered after major revisions

rejected

**Were a revised manuscript to be sent for another round of reviews:**

I would be willing to review the revised manuscript.

I would not be willing to review the revised manuscript.

**Suggestions for revision or reasons for rejection**

**(visible to the public if the article is accepted and published)**

The revised manuscript "Gravity waves above the Northern Atlantic and Europe during streamer events using Aeolus" by Wuest et al. reads much better now. My comments have been adequately addressed.

Still, there are a few corrections that should be made before publication in AMT. My main point is that there are still a few occurrences in the text that suggest Ekin would be a lower limit, which however is not clear.

Please find my specific comments below.

Thank you for your comments. I addressed them below and changed the manuscript accordingly.

- Two additional remarks: According to the request of AMT, I revised the color scheme of figure 7 and 8, I did not change the content. ٠

  - In lines 471 472 (version with marked changes) there is written "Aeolus measures a horizontal line-of-sight velocity; this means that it is particularly sensitive to GWs whose air parcels oscillate horizontally (inertia GW) and parallel to the line of sight." I changed the term in brackets from inertia to inertia and mid-frequency GW as also mid-frequency GW show a horizontal direction of oscillation.

**SPECIFIC COMMENTS:**

(1) I.16/17: Since the abstract is a very prominent place in the manuscript, you should avoid the uncommented use of "lower limit" because later in the manuscript it turns out that this expectation does not necessarily hold. Suggestion:

... this is strictly speaking a lower limit for the kinetic energy density. ->

... assuming a perfect instrument performance, this would be a lower limit for the kinetic energy density. Thanks for the suggestions, I included it.

(2) I.100: the region -> a prominent region Changed

(3) I.142: Just a comment.

My previous reviewer comment about the streamer structure was just a misunderstanding. What I was trying to say is that meridional transport of air causes zonal variations in the TO3 distribution. I think your text in the revised manuscript is OK as is. Thanks for the clarification.

(4) I.213: height -> magnitude Indeed!

(5) 1.343: later -> latter Done

(6) 1.393: 250 -> 250 hPa Done

(7) I.393: please resolve "GFS" Done

(8) I.472: I do not understand the following statement:

"The larger the zonal wavelength and the smaller the meridional one, the closer the measured amplitude comes to the true amplitude."

Just before you were talking about Aeolus limitations regarding short meridional wavelengths. This would suggest that Aeolus would be more sensitive to long meridional wavelengths

With this sentence I wanted to say that not only the averaging along the flight direction plays a role for detecting a wave and measuring its true amplitude but also the orientation of the wave: if the wave is oriented perpendicular to the line of sight, the averaging effect is much larger compared to a parallel orientation. As Aeolus looks to the side (35° off nadir), waves with longer zonal than meridional wavelengths are less attenuated. However, since this sentence is confusing and the main message is already given two sentences before the one we are discussing, I deleted it.

(9) I.475: bias -> high-bias Done

(10) I.508: please remove "lower bound" from the caption of Fig. 10 because Ekin may be severely high-biased due to measurement noise. I deleted the whole figure due to the comments of reviewer 2

(11) Caption of Fig.11, second line: left part -> right part Corrected

(12) 1.520: please avoid the expression "lower bound" because Ekin may be high-biased I replaced the sentence in brackets with "estimates

**Report #2**

Submitted on 27 Jun 2024 Anonymous referee #1

Anonymous during peer-review: Yes No

**Anonymous in acknowledgements of published article: Yes No**

**Checklist for reviewers**

| <ol> <li>Scientific significance</li> <li>Does the manuscript represent a substantial contribution to scientific progress within the scope of this
journal (substantial new concepts, ideas, methods, or data)?</li> </ol>               | Excellent Good Fair Poor |
|----------------------------------------------------------------------------------------------------------------------------------------------------------------------------------------------------------------------------------------------|--------------------------|
| 2) Scientific quality
Are the scientific approaches and applied methods valid? Are the results discussed in an appropriate and
balanced way (consideration of related work, including appropriate references)? Note that papers do not | Excellent Good Fair Poor |
| necessarily need to be long to be scientifically sound.                                                                                                                                                                                      | Excellent Good Fair Poor |
| 3) Presentation quality
Are the scientific results and conclusions presented in a clear, concise, and well structured way (number and                                                                                                     |                          |

**For final publication, the manuscript should be**

quality of figures/tables, appropriate use of English language)?

**accepted as is**

accepted subject to technical corrections

accepted subject to minor revisions

reconsidered after major revisions

rejected

Were a revised manuscript to be sent for another round of reviews: I would be willing to review the revised manuscript. I would not be willing to review the revised manuscript.

**Suggestions for revision or reasons for rejection**

(visible to the public if the article is accepted and published) SHORT SUMMARY The paper titled "Gravity waves above the Northern Atlantic and Europe during streamer events using Aeolus" has been revised somewhat to address points raised by this and other reviewers' comments. It has been improved, however there are still a number of issues that this reviewer has with the manuscript in its current form. These would need addressing before publication by AMT can be recommended by this reviewer.

**Thank you for your comments. I addressed them below and changed the manuscript accordingly. Two additional remarks:**

According to the request of AMT, I revised the color scheme of figure 7 and 8, I did not change the content.

In lines 471 – 472 (version with marked changes) there is written "Aeolus measures a horizontal line-of-sight velocity; this means that it is particularly sensitive to GWs whose air parcels oscillate horizontally (inertia GW) and parallel to the line of sight." I changed the term in brackets from inertia to inertia and mid-frequency GW as also mid-frequency GW show a horizontal direction of oscillation.

**GENERAL COMMENTS**

1) This reviewer still has concerns about the method used to extract GW perturbations and there is no substantial development of the analysis technique or a wider analysis which might help to diminish these concerns. The authors have instead sought to clarify in the discussion the reasons for their previous method choice, which they do by considering the tropopause as the main non-GW perturbation source, and by using the 'repeating spline' method to avoid 'strong deflections'. Although the reviewer would prefer to see a consideration for other important wind perturbation sources in the troposphere, the authors do now clearly discuss this limitation of their method. Indeed, the authors have added some caveats into the text to clarify the shortfalls of the GW extraction method as a consequence of the techniques they are using.

The analysis technique of adapting a spline to the data is common for the analysis of gravity waves (see, for example, the citations in the introduction of Wüst et al. (2017)). In order to avoid exactly this discussion about the repeating spline approach, I published a paper (Wüst et al, 2017) where I introduced it, compared it to the "normal" spline method and applied it to measured data. Even though there are other techniques than the adaption of a spline to extract gravity wave signatures from vertical measurement profiles (local or global horizontal filters, if enough data are available in the horizontal, for example, or vertical filters such as the Butterworth filter, which is a high-pass fast Fourier transform filter), there is no agreement in the community, which technique is the most appropriate one, or a publication known to me, where the optimal technique is published; every technique has its advantages and disadvantages. For the spline technique, the greatest challenge in this work is the tropopause; we discussed this challenge and split the analysis into tropospheric and stratospheric part.

Comparing the different approaches for the extraction of gravity waves from vertical measurement profiles is not the scope of this manuscript. Therefore, we do not use further analysis techniques here.

2) The focus of the study has not been widened much in response to the previous comments by all reviewers. It seems a little unusual to focus on just streamer events (and one specific type at that – see author comment on P05.L107 on page 5 of their response to RC2) and only using Aeolus, for an AMT paper, which is why the suggestion to widen this focus was made previously. Would this article be better suited to a diWerent journal if it is to be so focused, as the authors have acknowledged that it is?

This manuscript represents a significant part of the results achieved in an ESA project. The project belonged to a call which focused on innovative applications of Aeolus data, so on other ones than the originally planned. The AMT homepage says that this journal is dedicated to the "discussion of advances in remote sensing, as well as in situ and laboratory measurement techniques for the constituents and properties of the Earth's atmosphere." We tried to use the Aeolus data for the investigation of a possible gravity wave source. This source was not explicitly addressed before, nevertheless it can be expected that gravity waves are generated by these streamer events. Additionally, the purpose of Aeolus is not the investigation of gravity waves. So, from our point of view and probably also from ESA's point of view, this study discusses an innovative aspect of an advanced remote sensing instrument. That is why we think that this manuscript fits into the scope of AMT. Of course, it would be of interest to widen the study, however, we have here practical limits (time, budget, obligations in new projects).

3) Further changes are necessary to some of the figures before they are of publishable quality. Notably, figure 10 is very diWicult to understand with the filled contour method currently used to plot the data. Although the authors have clarified the dates corresponding to each week and

have defended their colour scale choice, they have not changed the plotting technique to make it easier to read. Furthermore, it is very diWicult to follow the story being told by this figure, in tandem with the corresponding text.

I removed figure 10. Finally, this figure does not provide much info – the most important info is: the amount of data is not enough to derive a temporal resolution of less than one week and this is to low for observing a phenomenon which lasts only some days. The manuscript is adapted accordingly.

SPECIFIC COMMENTS PXX.LXX Comment

P03.L63 However, for parts of the GW spectrum, equation (3)... (Please insert a comma after 'spectrum' to avoid ambiguity.) Done

P03.L87 The phrase "especially the precision, since it is not removed through the detrending procedure in contrast to a bias" is now a little confusing. Why is the precision being removed? Yes, you are right, I re-formulated it. In the same sentence a saw a mistake, which I also corrected: "the error ..., which is lower than originally expected"  $\rightarrow$  "the error ..., which is higher than originally expected" (the precision is lower, but not the error).

For Aeolus, which is more challenging, the precision or the accuracy?

Sorry, I don't understand the question. Challenging in which context? In the context of deriving the error or handling it, for the analysis of the Aeolus data, ... ? For Aeolus data, the error is not separated according to bias and precision. Validation campaigns mentioned in the manuscript provided hints, that the random error, so the precision, is higher than the bias. More info about the source of the random error are now provided in the text according to the comments of another reviewer (I. 240 ff. in version with marked changes).

P08.L178 This now reads a bit better, the authors have clarified that the variable being added is only being corrected for the satellite's orbital node and it is clearer. Thank you.

**FINAL COMMENT**

Overall, no significant changes have been made to the scientific method, scientific quality or presentation quality for this manuscript, and so it is not yet possible to recommend that the

article be accepted for publication without some of the changes outlined above and previously.

There have been some good alterations to the text which add important clarifications, however further revisions to the method and quality are still required for this reviewer to accept the paper.

**Referee Report:**

amt-2024-18-referee-report.pdf

**Report #3**

Submitted on 04 Sep 2024 Anonymous referee #4

Anonymous during peer-review: Yes No

Anonymous in acknowledgements of published article: Yes No

**Checklist for reviewers**

| <ol> <li>Scientific significance</li> <li>Does the manuscript represent a substantial contribution to scientific progress within the scope of this journal (substantial new concepts, ideas, methods, or data)?</li> </ol>                                                                           | Excellent Good Fair Poor |
|------------------------------------------------------------------------------------------------------------------------------------------------------------------------------------------------------------------------------------------------------------------------------------------------------|--------------------------|
| 2) Scientific quality
Are the scientific approaches and applied methods valid? Are the results discussed in an appropriate and balanced way (consideration of
related work, including appropriate references)? Note that papers do not necessarily need to be long to be scientifically sound. | Excellent Good Fair Poor |
| 3) Presentation quality
Are the scientific results and conclusions presented in a clear, concise, and well structured way (number and quality of figures/tables,
appropriate use of English language)?                                                                                         | Excellent Good Fair Poor |

**For final publication, the manuscript should be**

accepted as is

accepted subject to **technical corrections** accepted subject to **minor revisions reconsidered after major revisions**

rejected

**Were a revised manuscript to be sent for another round of reviews:**

I would be willing to review the revised manuscript.

I would not be willing to review the revised manuscript.

**Suggestions for revision or reasons for rejection**

**(visible to the public if the article is accepted and published)**

The revised manuscript "Gravity waves above the Northern Atlantic and Europe during streamer events using Aeolus" by Wüst et al. discusses the use of Aeolus data to derive the kinetic energy density over the Northern Atlantic and Europe during streamer events. These events are typically associated with increased gravity wave energy densities. A key challenge is the high noise level in Aeolus data, complicating the detection of gravity wave signals. To mitigate this, the authors average the data over a larger region, reducing uncertainty in the daily time series and enabling a more accurate correlation with streamer events. The study finds minor enhancements in kinetic energy density that may be linked to streamers, with horizontal distributions indicating other potential gravity wave sources as well.

As noted by the referees, the paper aligns well with the scope of AMT and offers valuable insights for the atmospheric dynamics community, especially for those focused on deriving gravity wave distributions from Aeolus measurements. In the revised manuscript, the authors have addressed most of the points raised by the referees and made the necessary changes to the text. However, several issues still need to be resolved before publication.

**Thank you for your comments. I addressed them below and changed the manuscript accordingly. Two additional remarks:**

- According to the request of AMT, I revised the color scheme of figure 7 and 8, I did not change the content.
  - In lines 471 472 (version with marked changes) there is written "Aeolus measures a horizontal line-of-sight velocity; this means that it is particularly sensitive to GWs whose air parcels oscillate horizontally (inertia GW) and parallel to the line of sight." I changed the term in brackets from inertia to inertia and mid-frequency GW as also mid-frequency GW show a horizontal direction of oscillation.

**General comment:**

Although the averaging approach to resolve the streamer events despite the high noise level in the Aeolus data is interesting, I wonder if the authors would have detected the two events without prior knowledge of the total ozone column measurements. When examining the timelines in Figs. 8 and 9, the enhancements in Mean E\_kin are not as pronounced as described in the text and are only noticeable when one knows what to look for. Therefore, the authors should clarify the primary objective of this study already in the introduction—namely, analyzing streamer events in terms of kinetic energy density using Aeolus data after they have been detected by other means. This clarification will help avoid potential confusion or disappointment for the reader who might expect the identification of gravity waves and/or streamers based on Aeolus data, given the title of the paper.

**Yes, that's true and I added this info in the abstract, intro and summary.**

Additionally, the unique benefits of using Aeolus data compared to other satellite data for streamer analysis should be more thoroughly elaborated in the text. I added some info in the paragraph starting in line 99 (version with marked changes).

**Specific Comments:**

1. The term "Atmospheric Dynamics Mission" (L. 15) and its acronym "ADM" (L. 17) are still used in the abstract and should be removed. Done, I also deleted it from table 1 (ADM-Aeolus Level-2B Algorithm Theoretical Baseline Document).

2. L. 204: Could the authors please clarify why Mie-cloudy winds, which have a smaller random error, were not used? I suspect this may be due to poor data coverage across the vertical profile. However, it might be possible to replace some of the noisy Rayleigh-clear winds with Mie-cloudy winds if they are available in the same region and altitude.

The L2B winds are available as Rayleigh-clear, Rayleigh-cloudy, Mie-clear and Mie-cloudy. The best quality has Rayleigh-clear and Mie-cloudy winds (Rennie et al. 2021). Mie-cloudy winds have smaller random errors (with a standard deviation of about 2.8–3.6 ms-1) than Rayleigh-clear winds (4.0–7.0 ms-1) winds (Rennie et al. 2021). However, there exist more Rayleigh-clear wind data than Mie-cloudy data (see figure 1 of Rennie et al. 2021). Additionally, Mie-cloudy data is hardly available in the upper part of the vertical profile, as clouds are very rare above the Northern Atlantic upper troposphere where the atmosphere is stably stratified and therefore suitable for gravity wave analyses. Therefore, we decided to use Rayleigh-clear data for the analysis when starting the project. I inserted this info also in the manuscript (l. 217 ff., version with changes marked).

We now checked whether it would be an option to merge both data sets. Both data products have different horizontal resolutions. Since March 2019, Rayleigh-clear data is averaged over 87 km, whereas Mie-cloudy are averaged over 12 km (Rennie et al. 2021). That is not ideal since it would make the analysis slightly inconsistent but would may be worth the effort, if enough profiles are available. We used the VIRES tool to for the time period of the first streamer (Nov. 1st – 8th, 2020). You find the screenshots at the end of the document. As you can see, Mie-cloudy winds are hardly available above the tropopause. So, we conclude that the Mie-cloudy winds are not helpful in this context.

3. L. 211 ff.: Relevant information on the altitude-dependence of the Aeolus wind error can be found in the following publications by ECMWF:

- Rennie, M.P., Isaksen, L., Weiler, F., de Kloe, J., Kanitz, T. & Reitebuch, O. (2021): The impact of Aeolus wind retrievals on ECMWF global weather forecasts. Q. J. R. Meteorol. Soc., 147, 3555–3586. https://doi.org/10.1002/qj.4142

- Rennie, M.P. & Isaksen, L. (2024): The NWP impact of Aeolus Level-2B winds at ECMWF, ESA Contract Report, 02/2024. https://www.ecmwf.int/en/elibrary/81546-nwp- impact-aeolus-level-2b-winds-ecmwf

I suggest adding a short paragraph on the wind error performance, focusing specifically on the Rayleigh-clear winds during the period of the two case studies. I combined this with the results of the next comment and inserted a paragraph starting in line 240 (version with marked changes).

4. Line 230: The restriction to profiles that cover at least 21 height bins seems quite strict and might be relaxed to enlarge the dataset without compromising accuracy, especially since a spline fit is applied to the profiles. A few missing data points within the profile may be acceptable under this approach.

**Yes, that's true and I checked it. For my analysis, I used the same data as for figure 4 (so Nov 2020, 25 - 70 °N, 0 - 20°E).**

Since we are looking for gravity waves, it is important that the stratosphere, which is stably stratified, is covered by the data. As stratospheric interval 11 km and higher is used in the manuscript. This corresponds approximately to level 1 - 9 which cover 18.6 - 10.7 km. Nine data points are not much and since we analyzed vertical wavelengths from 5 to 10 km, the profile should at least cover those nine height bins.

I then checked how many profiles of the data basis mentioned above cover the bins 1 - 9 and the bins 1 - 21. In the first case, I get 2412 profiles, in the second 2216. So, analyzing additional profiles that cover the absolute minimum of data points would give me 9% more profiles or in other words, profiles that cover the bin range 1 - 9 cover also the bins 1 - 21 in the absolute majority of cases. Therefore, adding this comparatively very low number of profiles will very likely not change the results.

In this context, I found an error in the text: I. 261 (version with marked changes) – there was written level 1 is the lowest altitude, it is the highest. I changed it.

5. Fig. 3: Adding the measurement tracks of Aeolus to the map would be beneficial, as it would provide a clearer impression of the coverage of Aeolus wind data in the region of interest. Done

6. I assume that the three maxima for the Rayleigh-clear HLOS wind error shown in Fig. 4 are related to differences in the thickness of the vertical range bins, which in turn affect the signal-to-noise ratio and consequently the wind error. Could the authors please verify this assumption?

That's an interesting idea, thank you. I checked it, however, it does not seem to hold, at least for this part of the data set.

For my analysis, I used the same data as for figure 4 (so Nov 2020, 25 – 70 °N, 0 – 20°E). I calculated the mean height of a bin, the difference between the mean bin heights and the mean error per bin for the bins 1 – 21. Next to this text, you find the results.

The mean error depends most clearly on the height (lowest plot): it varies around 4 m/s for ca. 6 km height and higher, below it can reach up to 13 m/s. The thickness of a bin (so the difference between the mean bin heights) shows a minimum around the tropopause and maxima in the upper part of the profile and around 4 km height (middle plot). The mean error can be relatively low for all bin thicknesses (upper plot, error values around 4 m/s for mean bin heights between ca. 550 m and 1150 m).

7. L. 428f.: Please rephrase this sentence: "They could be the results of the relatively high Aeolus error, specifically the if it is due to a low precision." Changed into "They could be the results of the relatively high Aeolus error, specifically if the error is due to a low precision."

8. L. 496f.: Please rephrase this sentence: "Figure 11a) and b) depict the height of the first and the first as well as the second maximum per Aeolus wind profile." Done

**Technical Corrections:**

1. Throughout the manuscript, units should be written in exponential form (e.g., m  $s^{-1}$ , J k $g^{-1}$ ) to comply with the AMT style guidelines. Changed in the text and in figures 4, 7, 8, 9, and 10

2. L. 343: "latter solution" instead of "later solution" Done

3. L. 393: "250 hPa" instead of "250" Done

4. Caption of Fig. 11: "... the right part includes also the secondary maximum". Done

Addressing these points will strengthen the manuscript and ensure it meets the high standards expected for publication in this field.

---

## Author Response (AR3)

Thank you for the comments. I changed the manuscript accordingly.

**Reviewer comments:**

**Report #1**

I changed "Rayleigh clear and Mie cloudy winds have the best quality of the four different wind products. Even though Mie cloudy winds have smaller random errors than Rayleigh clear winds (Rennie et al. 2021), they are rare above the upper troposphere where the atmosphere is stably stratified and therefore suitable for gravity wave analyses." to "Rayleigh clear and Mie cloudy winds have smaller random errors than Rayleigh clear winds (Rennie et al. 2021), they of the four different wind products. Even though Mie cloudy winds have smaller random errors than Rayleigh clear winds (Rennie et al. 2021), they are rare above the upper troposphere. Especially the stratosphere is very suitable for gravity waves analysis as it is stably stratified. Therefore, Rayleigh clear winds are preferable to Mie cloudy winds for gravity wave analysis."

**Report #2**

I changed

"to high" to "too high" (formerly I. 102) and

"In this case, Aeolus is also superior to ground-based and balloon-based wind measurements, as these are generally not carried out over the Atlantic and therefore not in the geographical area where streamers form." to "In this case, Aeolus is also superior to ground-based and balloon-based wind measurements, as these are generally not carried out over the Atlantic, one of the regions where streamers preferentially form." (formerly I. 105).

Reading the whole manuscript again, I found some further technical issues, which I corrected (line numbers refer to version with marked changes):

Figure 2 caption: westward  $\rightarrow$  eastward (the streamer moved eastward in time, as mentioned for case 1 in figure 1, too)

I. 226: inserted "wind" after hlos

I. 246: in  $\rightarrow$  is

I. 299: removed "generated"

I. 448: Figure 2  $\rightarrow$  Figure 3

I. 488: the  $\rightarrow$  this

I. 518: E\_kin  $\rightarrow E_{kin}$

And finally, I added the reviewers to the acknowledgement.